# Reanalysis of ribosome profiling datasets reveals a function of rocaglamide A in perturbing the dynamics of translation elongation via eIF4A

Fajin Li [1,2,3,4] ✉, Jianhuo Fang[1,2,4], Yifan Yu[1,2,4], Sijia Hao[1,2,3], Qin Zou[1,2], Qinglin Zeng[1,2] & Xuerui Yang [1,2,3] ✉

The quickly accumulating ribosome profiling data is an insightful resource for studying the critical details of translation regulation under various biological contexts. Rocaglamide A (RocA), an antitumor heterotricyclic natural compound, has been shown to inhibit translation initiation of a large group of mRNA species by clamping eIF4A onto poly-purine motifs in the 5′ UTRs. However, reanalysis of previous ribosome profiling datasets reveals an unexpected shift of the ribosome occupancy pattern, upon RocA treatment in various types of cells, during early translation elongation for a specific group of mRNA transcripts without poly-purine motifs over-represented in their 5′ UTRs. Such perturbation of translation elongation dynamics can be attributed to the blockage of translating ribosomes due to the binding of eIF4A to the poly-purine sequence in coding regions. In summary, our study presents the complete dual modes of RocA in blocking translation initiation and elongation, which underlie the potent antitumor effect of RocA.

Although the initiation of translation has been well recognized as a rate-limiting step of protein synthesis[1], quickly accumulating evidence also illustrates crucial regulatory machineries during translation elongation controlling the rate and quality of protein synthesis[2–4]. Stress conditions such as heat shock, oxidative stress, amino acid starvation, ultraviolet irradiation (UV) and drug treatment induce ribosome stalling during early translation elongation[5–10], which may cause ribosome collisions and inhibit protein synthesis[9]. In addition, ribosomes tend to move slowly at some specific positions, such as polyproline motifs[11–14], suboptimal codons[3,15], and codons for amino acids with high positive charges[3]. The availability of tRNAs, translational co-folding and mis-folding of the nascent peptide chain are also associated with variations in the translation elongation rate[3,6,7,16]. More recent studies also showed that loss of translation factors and ribosome proteins per se would also induce

ribosomes stalled at the early elongation stage and reduce translation efficiency[4,11,17,18].

Via deep sequencing of ribosome-protected RNA fragments (RPFs), ribosome profiling has revealed the landscape of ribosome occupancy on ORFs at the genome-wide scale and sub-codon resolution[19–21]. Based on the assumptions that translation initiation is the rate-limiting step and that the overall translation elongation rate of a particular gene is consistent across different conditions, the total RPF count of a gene has been frequently used as a proxy of the relative rate of protein synthesis[1,20,22]. This principle holds in most previous studies, which have revealed novel insights into the translational regulation machineries, mostly in the stage of translation initiation, under a large variety of biological contexts. However, given the complicated and extensive factors potentially driving dynamic shifts of translation elongation, we believe that it would be

[1]MOE Key Laboratory of Bioinformatics, School of Life Sciences, Tsinghua University, Beijing 100084, China. [2]Center for Synthetic & Systems Biology, Tsinghua University, Beijing 100084, China. [3]Joint Graduate Program of Peking-Tsinghua-National Institute of Biological Science, Tsinghua University, Beijing 100084, China. [4]These authors contributed equally: Fajin Li, Jianhuo Fang, Yifan Yu. ✉e-mail: lfj17@tsinghua.org.cn; yangxuerui@tsinghua.edu.cn

of unique value to revisit previous studies with ribosome profiling and reevaluate the presumption of translation elongation being undisturbed for specific genes.

In fact, distribution patterns of the RPFs along the ORFs have been frequently shown to be highly associated with the dynamics of translation elongation[4,11,19]. Over-representation of the RPFs at specific sites along the ORF usually indicates frequent ribosome stalling or ribosome pausing, which should lead to a reduced, rather than increased, rate of translation[3,4,15,21,23]. Therefore, such polarized changes of the RPF distributions should be a strong signature of translation elongation perturbation that may have invalidated the above presumptions.

Rocaglates, for example rocaglamide A (RocA), are a class of natural products isolated from the plant genus *Aglaia*, which are characterized by a common cyclopenta[b]benzofuran skeleton[24,25]. RocA and many of its derivatives have been demonstrated to be potent antitumor drugs in hematological and solid tumor cancers[24,26–31]. Interestingly, some rocaglates can kill tumor cells while showing no or little toxicity to primary T or B cells and heathy spleen and liver tissues in mouse models, thereby making it an attractive candidate for applications in clinical oncology[24,26,28,32–34]. Currently, several rocaglates are in preclinical[27,32,35–37] and clinical trials[38]. Recently, a synthetic rocaglate CR-31-B (-) was shown to inhibit SARS-CoV-2 replication at non-cytotoxic low nanomolar concentrations in vitro and ex vivo, suggesting the antiviral potential of the compound[39].

The antitumor function of RocA or its derivatives was mainly attributed to its potent inhibitory effect on protein synthesis[24]. Recent studies showed that RocA directly blocks the scanning of the 43S preinitiation complex (PIC) on a large group of mRNAs by clamping eIF4A and DDX3 onto the poly-purine sequences in the 5′ UTR, thereby inhibiting translation initiation[40–42]. This often leads to elevated uORF translation with abnormal 80S ribosome accumulation in the 5′ UTR[40]. However, given that RocA induced global translation inhibition[40], this mode of action did not fully explain how translation of the mRNA transcripts whose 5′ UTR do not contain poly-purine sequences can also be inhibited by RocA. A more recent study further proposed that by clamping eIF4A on poly-purine motifs, RocA reduced the abundance of free eIF4A that are available for the formation of eIF4F complexes for the other mRNA transcripts without poly-purine motifs, thereby leading to the inhibition of global translation initiation[43,44].

In the present study, we reevaluated the dynamics of translation under a large variety of biological contexts by revisiting 49 previously published ribosome profiling datasets in human cells. Such an analysis strategy allows systematic comparisons among different types of experimental conditions, which should generate insights into the overall machinery of translation perturbation for a given conditions. Interestingly, our meta-analysis revealed an unexpected function of RocA in shifting the ribosome occupancy patterns in the CDS regions during translation elongation. Further analyses showed that RocA selectively perturbed the early translation elongation of a set of genes, which are characterized by short 5′ UTRs without the enrichment of poly-purine motifs, suboptimal codons, and more positive charges and hydrophobicity in the N-terminal regions of their protein products. These genes are highly involved in fundamental mitochondrial processes such as ATP synthase and the mitochondrial respiratory chain complex. Our own disome-seq data and translation reporter assays confirmed that perturbation of translation elongation in response to RocA resulted in ribosome collision and reduction of protein synthesis.

Interestingly, integrative analyses of iCLIP-seq and ribosome profiling data further showed that the inhibition of translation elongation by RocA can also be attributed to the blockage of translating ribosomes due to the direct binding of eIF4A to the poly-purine sequence in the CDS regions. Therefore, we have uncovered dual modes of action of RocA via shifts of translation initiation and early elongation, which are repeatedly shown in a series of cancerous cells. Both functions were dependent on eIF4A but executed on different

sets of genes. In summary, the extraction of novel and critical insights from resources of previous ribosome profiling data has greatly helped us elucidate the complete and detailed machinery of RocA as a potent antitumor drug via its interference with the dynamics of translation.

## Results

### RocA induces shifts of ribosome distribution along the CDS during translation elongation

Previous studies have well demonstrated that RocA induces translation inhibition by clamping eIF4A on poly-purine sequences in 5′ UTRs and blocking translation scanning of 43S PIC[40,41,43]. Indeed, based on reanalysis of previously published ribosome profiling data of HEK293 cells (GSE70211)[40], the mRNAs with poly-purine motifs in their 5′ UTRs exhibited strong accumulation of ribosome footprints in the 5′ UTRs upon RocA treatment for 30 min (PTGES3 and EIF2S3 as two examples, Supplementary Fig. 1a), which is mainly attributed to the elevated uORF translation due to the RocA-induced blockage of 43S PIC scanning[40].

However, not all mRNA transcripts harbor poly-purine motifs in their 5′ UTRs. In addition, for the mRNAs whose translation initiation is scanning independent, it has been unclear whether and how their translation was shifted by RocA treatment. For example, translation initiator of short 5′ UTR (TISU), located within 5–30 nucleotides from the 5′ cap and with a median length of 12 nucleotides, could direct efficient cap-dependent translation initiation without eIF4A and scanning[45,46]. Unexpectedly, as revealed by the same ribosome profiling data of HEK293 cells mentioned above, treatment with RocA for 30 min induced strong enrichment of ribosome occupancy after the start codons of some mRNAs with TISU (Supplementary Fig. 1b), for example, UQCC2 and NDUFS6, of which the translation initiations have been demonstrated to be eIF4A-independent[47]. No ribosome stalling was observed in the 5′ UTRs of these mRNAs (Supplementary Fig. 1b). This finding indicates a previously uncharacterized novel function of RocA in the process of mRNA translation elongation, in addition to its well-acknowledged repressive effect on translation initiation.

To gain a more comprehensive and more precise view of the potential functions of RocA on translation elongation, we systematically compared the global shifts of translation dynamics in response to a large variety of experimental conditions. Specifically, we reanalyzed a series of published ribosome profiling datasets in human cells (Supplementary Data 1) and quantified the polarity scores of the mRNA transcripts under a large variety of biological contexts, including RocA treatments and many other conditions perturbing translation initiation and/or elongation. As a quantitative assessment of the balance of ribosome occupancy along the CDS from the 5′ to the 3′ ends, the polarity score indicates the dynamics of translation elongation for each mRNA transcript[11]. Therefore, differences in the polarity scores (Δpolarity) by comparing the conditions of translation perturbations to their control conditions indicate the shifted, if any, dynamics of translation elongation in response to the conditions of translation perturbations. These treatment conditions, 49 in total (Supplementary Data 1), were then subjected to hierarchical clustering based on the Δpolarity profiles of all the genes.

As expected, these 49 conditions were classified into two major clusters, one containing mostly the known inhibitors of translation initiation, such as hippuristanol, knockdown of eIF1, Torin-1[48–50], and the other one mainly composed of inhibitors of translation elongation, such as heat shock, proteotoxic stress, CHX treatment, UV treatment, knockdown of RPL12, et al.[6,7,9,18] (Fig. 1a). From a global perspective, in the first major cluster, the inhibitors of translation initiation led to only mild, if any, shifts of the polarity scores, which nicely echoes the note that initiation blockage does not affect the general patterns of ribosome occupancy along the CDS during translation elongation (Fig. 1a). In contrast, the cluster of translation elongation perturbations showed strong and predominantly negative shifts of the polarity

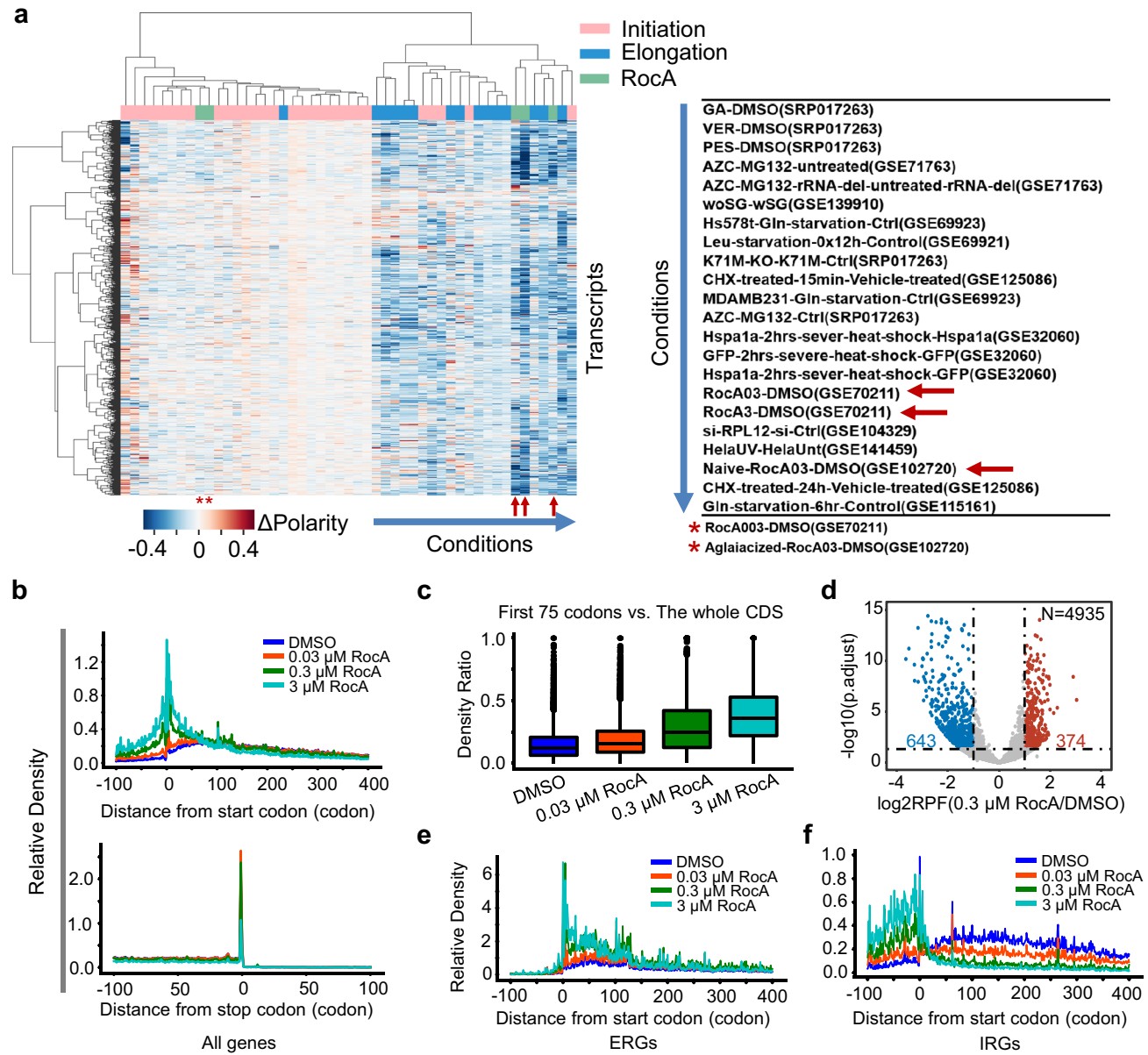

**Fig. 1 | Reanalysis of ribosome profiling datasets reveals patterns of ribosome occupancy. a** Heatmap (left panel) showing hierarchical clustering of 49 conditions of translation perturbation, based on the profiles of the polarity differences by comparing the treatment to the control groups. The 49 conditions were arranged by columns, and the 4842 transcripts were arranged by rows. The conditions of the second major cluster are listed to the right. The red arrows indicate the three RocA treatment conditions. The red asterisks indicate another two conditions of RocA. **b** Metagene plots of the averaged RPF read densities for all the transcripts (GSE70211). The X-axis represents the distance from the start codon (upper panel) and stop codon (lower panel). The transcripts used for the plot were

9712 (DMSO), 9429 (0.03 μM RocA), 8467 (0.3 μM RocA), and 8515 (3 μM RocA). **c** Boxplots showing the ratio of ribosome density at the first 75 codons over the whole CDS region, and the transcripts (*n* = 7007) used for analysis are the common ones for all samples in **b**. Center line, median; box limits, upper and lower quartiles; whiskers, 1.5x interquartile range; points, outliers. **d** Volcano plot showing the results of differential expression analysis with the RPF reads from the ribosome profiling data. The significant p values are generated via DESeq2 and adjusted with the Benjamini and Hochberg method by default. Metagene plots showing ribosome distribution of the ERGs (**e**) and IRGs (**f**). Source data are provided as a Source Data file.

scores (i.e., negative values of Δpolarity), indicating a global shift of ribosome occupancy toward the 5′ direction of the CDS (Fig. 1a).

Surprisingly, the middle (0.3 μM, RocA03) and high (3 μM, RocA3) concentrations of RocA in HEK293 cells were not categorized as expected into the cluster of translation initiation inhibitors but into the cluster of elongation inhibition (Fig. 1a). Specifically, similar to the other conditions known to repress translation elongation, RocA induced a globally negative shift in polarity, indicating unbalanced accumulation of ribosomes at earlier phases of translation elongation (Fig. 1a, Supplementary Fig. 2a). Notably, in HEK293 cells with eIF4A double mutations, which acquired resistance to the drug[41], RocA

(0.3 μM, RocA03) failed to suppress the polarity scores (Fig. 1a, Supplementary Fig. 2b). Together, the observations above indicate a potent yet previously uncharacterized perturbation of translation elongation induced by RocA, which may also be dependent on eIF4A.

**RocA functions in translation initiation and elongation in a gene-specific manner**

To further characterize the ribosome distribution patterns perturbed by RocA in more detail, we pooled and aligned all the ribosome-protected fragments (RPFs) based on the distances from their P-sites to the start or stop codons. Such metagene analyses of ribosome

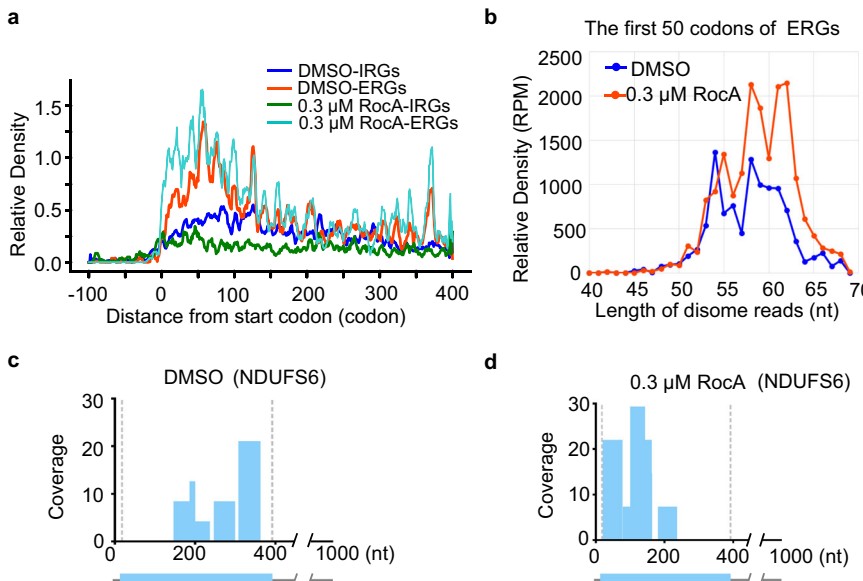

**Fig. 2 | Disome-seq reads showing the colliding ribosomes along transcripts upon RocA treatment. a** Metagene plot showing the disome distribution of ERGs and IRGs. **b** Length distributions of the disome-seq reads within the first 50 codons of ERGs. **c**, **d** Disome distribution along the transcript of NDUFS6 upon DMSO and RocA treatment. Source data are provided as a Source Data file.

occupancy showed strong accumulation of ribosomes in both the upstream and downstream regions around the start codons in response to RocA in a dose-dependent manner (Fig. 1b). These results confirmed the role of RocA as an inhibitor of translation initiation and further supported our discovery that RocA could perturb the dynamics of translation elongation as well. It appears that after the start codons, the increased accumulation of ribosomes induced by RocA mainly takes place within the first 75 codons during translation elongation (Fig. 1b). Indeed, the percentages of RPFs in the first 75 codons of the mRNA transcripts were dramatically increased by RocA (Fig. 1c). It remains to be answered whether such dual functions of RocA were executed in a gene-specific or global manner.

To explore the translation perturbation of each gene by RocA, we performed differential expression analysis with the RPFs mapped to the CDS of each gene, excluding the reads on the first 15 codons and the last 5 codons. As expected, upon RocA treatment (0.3 μM), the RPFs were downregulated for a large group of genes (Fig. 1d, Supplementary Data 2), which is a typical signature of translation initiation inhibition. This is well in line with previous reports that RocA selectively blocks translation scanning of 43S PIC in the 5′ UTRs of specific mRNAs, thereby suppressing translation initiation[40]. Therefore, we name these genes as initiation repressed genes (IRGs). However, there also existed a group of genes with upregulated RPFs (Fig. 1d, Supplementary Data 2), which could not be attributed to the previously acknowledged function of RocA as an inhibitor of translation initiation. Instead, as would be shown by the following analyses of this study, these genes were perturbed by RocA at the stage of translation elongation, thereby being name as elongation repressed genes (ERGs). Note that the ERGs and IRGs were highly consistent upon treatment with RocA at the middle (0.3 μM) and high (3 μM) concentrations (Supplementary Fig. 2c). Therefore, the ERGs and IRGs in response to the middle concentration of RocA (0.3 μM) would be used from this point on, unless explicitly stated otherwise.

Next, metagene analyses with the set of ERGs or the IRGs separately exhibited strikingly distinct patterns of ribosome occupancy shift in response to RocA (Fig. 1e, f, Supplementary Fig. 2d). Specifically, for the set of IRGs, RocA induced ribosome accumulation in 5′ UTRs and deprivation in the CDS, which is a typical pattern of translation initiation blockage. In contrast, such a pattern was almost

completely lost for the ERGs set. Instead, RocA induced dramatic accumulation of ribosomes in the CDS of the ERGs near the start codon in a dose-dependent manner, indicating stalling of early translation elongation (Fig. 1e, f, Supplementary Fig. 2d).

Taken together, the results above indicate two modes of action of RocA on translation in a gene-specific manner. The first typical mode takes place in the 5′ UTRs, resulting in initiation blockage of a set of genes as represented by the IRGs. The second mode of action functions on the CDS, leading to perturbed dynamics of translation elongation for a different set of genes as represented by the ERGs.

### Disome-seq revealed ribosome collisions induced by RocA during early translation elongation of the ERGs

The analyses above uncovered a function of RocA in inducing ribosome stalling during early elongation of the ERGs. Next, we performed disome-seq to assess the potential events of ribosome collisions[12–14,51] as a result of ribosome stalling.

As expected, due to the inhibition of translation initiation, the disome densities in the CDS regions of the IRGs were greatly reduced upon RocA treatment (Fig. 2a). In contrast, for the ERGs, the disome densities were increased in the first ~50 codons (Fig. 2a). Note that the apparently peculiar periodic distribution of disome in ERGs is an artifact due to data fluctuation, the process of data smoothing, and the low resolution of the X axis. In fact, autocorrelations of the disome densities in the CDSs of ERGs indicate no periodicity (data not shown). The increases of disome densities were seen for all the different lengths from 55 nt to 65 nt of the disomes found in the first 50 codons of the ERGs (Fig. 2b). Taking a typical ERG NDUFS6 as an example again, the disome-seq reads are clearly enriched within the first ~150 nt of the CDS region upon RocA treatment (Fig. 2c, d), which echoes ribosome stalling during early translation elongation, as illustrated in Supplementary Fig. 1b. Therefore, our disome-seq results confirmed the function of RocA in inducing ribosome collision due to ribosome stalling during early translation elongation.

### ERGs are characterized by features associated with suboptimal translation elongation

The results above have illustrated dual modes of action of RocA on translation initiation of the IRGs and early elongation of the ERGs.

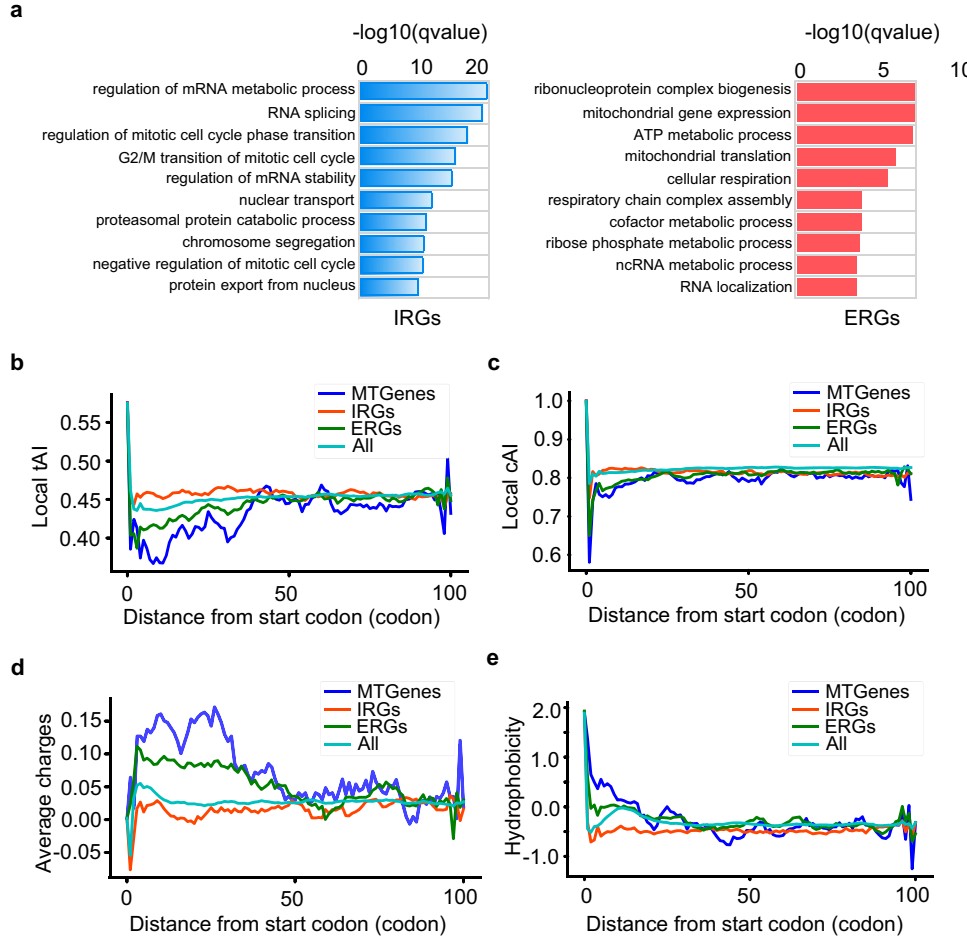

**Fig. 3 | Characterizations of ERGs and IRGs. a** The results of gene ontology (GO) analyses. **b** Metagene plots of local tAI for different gene sets, including a set of all the 19636 transcripts (All), 374 ERGs, 643 IRGs, and 64 mitochondrial function-associated genes (MTGenes). **c** Metagene plots of local cAI for the different gene sets presented in **b**. Average charges (**d**) and hydrophobicity (**e**) of the amino acids for the gene sets presented in panel b. Source data are provided as a Source Data file.

Specifically, the IRGs were enriched by the functions of mRNA metabolic process, RNA splicing, cell cycle, and nuclear transport functions (Fig. 3a, left panel), while the ERGs were significantly enriched by multiple mitochondrial processes, including mitochondrial gene expression, mitochondrial translation, and functions associated with cellular respiration (Fig. 3a, right panel). Indeed, upon RocA treatment, the 64 previously annotated mitochondrial function-associated genes (MTGenes) in ERGs showed a strong bias of ribosome occupancy toward the 5′ regions of the CDS (Supplementary Fig. 3a, b, Supplementary Data 3).

As expected, the IRGs were characterized by longer 5′ UTRs enriched by poly-purine motifs and more complex secondary structures than the ERGs and the full transcriptome (Supplementary Fig. 4a–c). This is consistent with the features of the previously identified RocA-sensitive genes[40]. In contrast, the ERGs had shorter 5′ UTRs with no enrichment of poly-purine motifs (Supplementary Fig. 4b). Note that there was a large overlap between the ERGs (22.2%) and the mRNAs with TISU elements[52] (Supplementary Fig. 4d), which echoes our previous observation that RocA induced ribosome accumulation in 5′ CDSs of the mRNAs with TISU elements (Supplementary Fig. 1b).

Furthermore, compared to the IRGs, the MTGenes and ERGs had lower local tRNA adaptation index (tAI) and codon adaptation index (CAI) in the regions of early elongation approximately within the first 50 codons (Fig. 3b, c), indicating relatively lower efficiencies of translation elongation. The N-terminal regions of the proteins encoded by the ERGs and MTGenes were enriched by amino acids with positive charges, such as arginine and lysine (Fig. 3d, Supplementary Fig. 5a, b), whereas the N-terminal regions encoded by the IRGs were enriched by amino acids with negative charges, such as glutamic acid and aspartic acid (Supplementary Fig. 5c). In addition, the N-terminal (~30–40 amino acids) of the nascent chains encoded by the ERGs were more hydrophobic than those encoded by the IRGs (Fig. 3e). It was reported that ribosomes move more slowly at the codons of positively charged[3,53] and hydrophobic[4,6] amino acids. Therefore, our results suggest that the ERG transcripts in general are subjected to suboptimal translation with slow speed during early elongation, which was further suppressed by RocA, thereby resulting in ribosome collision and accumulation. However, this does not fully explain why the ERGs and the IRGs respond to RocA differently.

## Inhibition of early elongation induced by RocA depends on eIF4A

It has been well characterized that RocA promotes clapping of eIF4A on poly-purine sequences in 5′ UTRs, which blocks the scanning of the 43S preinitiation complex (PIC) and thereby inhibits translation initiation. This machinery was shown to be cap-/eIF4F-/ATP-independent[40,41]. As mentioned previously, the 5′ UTRs of the IRGs are indeed enriched by poly-purine motifs (Supplementary Fig. 4a). Interestingly, although there was no enrichment of poly-purine motifs in the 5′ UTRs of the ERGs, they were frequently found and enriched in the CDS regions (Fig. 4a). It is a plausible hypothesis that eIF4A could also bind to the CDS of the ERGs on these poly-purine motifs in the presence of RocA.

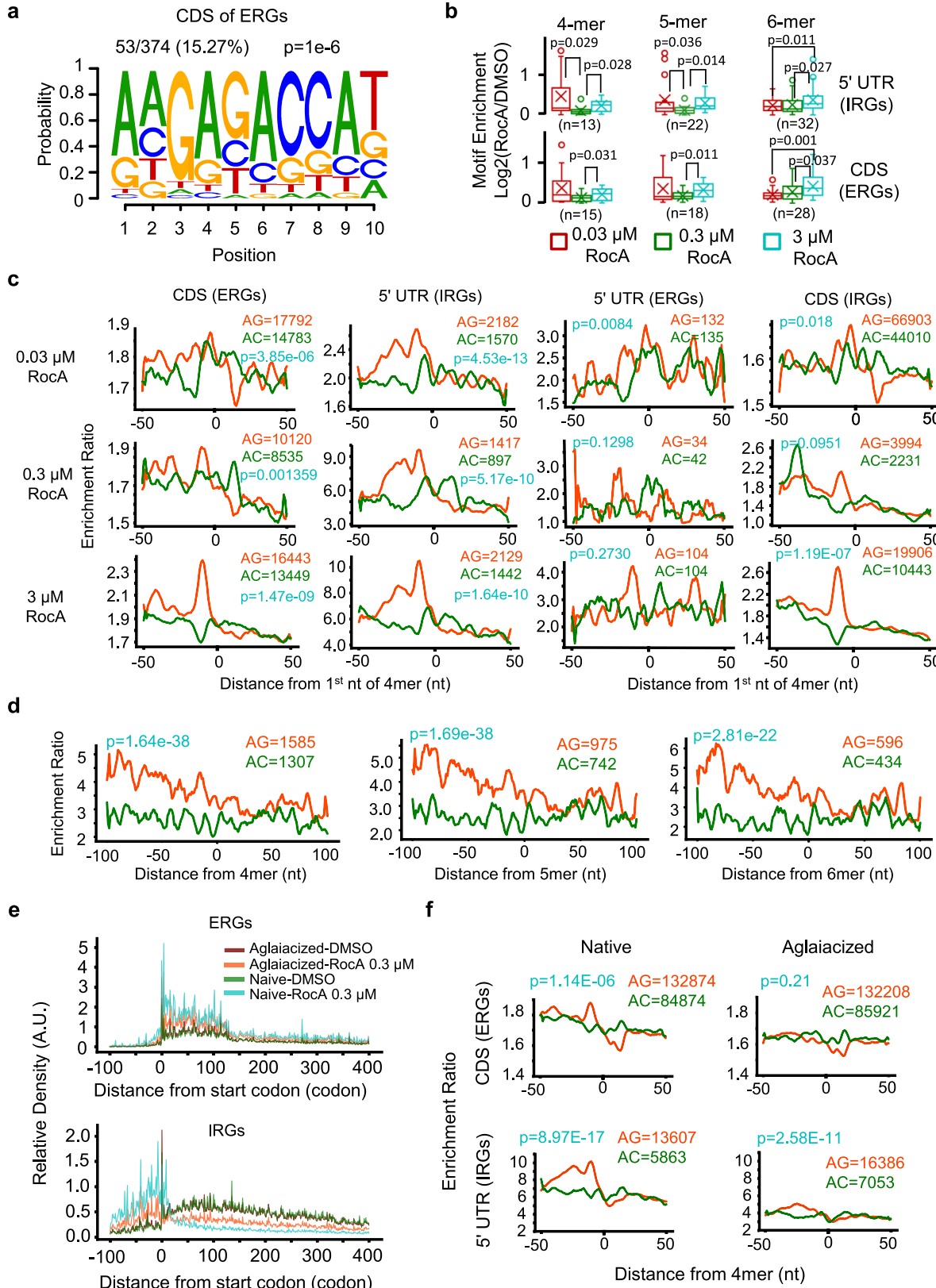

Here we designed plasmid constructs of the ERG CDS regions fused with MS2-binding sites, which were transfected into the cells expressing MS2-GFP (Supplementary Fig. 6a). The MS2-GFP-RIP assays with the CDS regions of 3 ERGs showed coimmunoprecipitation of eIF4A with GFP, suggesting binding of eIF4A with the CDS-MS2bs sequences (Supplementary Fig. 6b). Importantly, treatment of RocA at 0.3 μM resulted in further increase of the binding affinity (Supplementary

Fig. 6b). Indeed, reanalysis of previously published eIF4A iCLIP-seq data[40] indicated that short poly-purine motifs (4–6 nt) bound by eIF4A were enriched in the 5′ UTRs of IRGs and in CDSs of ERGs, both of which were further enhanced by RocA treatments (Fig. 4b).

We then asked whether the poly-purine motifs in the CDSs of ERGs are associated with ribosome stalling during translation elongation. Specifically, with published ribosome profiling datasets[40], we calculated

**Fig. 4 | RocA inhibits translation elongation via the direct binding of eIF4A1 to the polypurine motifs. a** Poly-purine motifs enriched in CDS regions of ERGs. **b** Enrichment of poly-purine motifs revealed by iCLIP-seq data for both the 5′ UTR of IRGs and the CDS of ERGs. The 4-mer, 5-mer, and 6-mer represent poly-purine motifs with 4, 5, and 6 nucleotides. Significance was calculated by two-sided Student's *t* test. "*n*" represents the number of kmer motifs used for statistics. Center line, median; box limits, upper and lower quartiles; whiskers, 1.5x interquartile range; points, outliers; "x", mean. **c** Enrichment ratio of ribosome density around poly-purine motifs (4mer) based on comparisons between RocAs and DMSO.

**d** Enrichment ratio of disome density around poly-purine motifs in the CDS regions of ERGs upon 0.3 μM RocA treatment. **e** Metagene plot of the ribosome densities for ERGs and IRGs in normal HEK293 cells and cells with EIF4A1 double mutations (Phe163Leu-Ile199Met). Naive: normal HEK293 cells. Aglaiacized: HEK293 cells with EIF4A1 double mutations. **f** Enrichment ratio of ribosome density around poly-purine motifs (4mer) based on comparisons between RocAs and DMSO in HEK293 cells with EIF4A1 double mutations (Aglaiacized). *P*-values in (**c**, **d**) and (**f**) are all calculated by two-sided Student's *t* test without adjustments. Source data are provided as a Source Data file.

RPF densities around the poly-purine motifs. As shown in Fig. 4c and Supplementary Fig. 7, RocA treatment induced strong upregulation of the ribosome densities in the upstream regions of the poly-purine motifs (poly-AG in 4-, 5-, and 6-mers), not only in the 5′ UTRs of IRGs but also in the CDS regions of ERGs. Such patterns of ribosome accumulation in the upstream regions only take place for the poly-purine motifs but not for the poly-AC motifs as negative references[40,42] (Fig. 4c and Supplementary Fig. 7). In addition, the disome-seq data also revealed that ribosome collisions induced by RocA tended to occur upstream of the poly-purine motifs rather than poly-AC (Fig. 4d).

There is no significant accumulation of ribosome footprints around the polypurine motifs in the 5′ UTR of ERGs upon RocA treatments (Fig. 4c), possibly because of the short 5′ UTRs. In fact, there are very few poly-purine motifs found in the ERG 5′ UTRs for the analysis above. By contrast, RocA induced significant ribosome stalling in the upstream of the polypurine motifs of the CDS of IRGs (Fig. 4c), but too much weaker extents compared to the ribosome stalling in the 5′ UTR of the IRGs. Indeed, although it is well acknowledged that RocA clamps eIF4A onto poly-purine motifs in the 5′ UTR[40], this machinery would certainly not completely shut down translation initiation of all the mRNA transcripts of IRGs in every single-cell. Thus, the ribosomes that successfully passed the start codons could be also subjected to blockage due to clamping of eIF4A on the CDS regions, like what we have observations for ERGs.

Next, we compared the relative enrichment of the polypurine sequences (≥4mer) in the first 75 codons and the remaining downstream CDS sequences (Supplementary Fig. 8a). Interestingly, the polypurine sequences are just slightly more enriched in the regions after the first 75 codons (Supplementary Fig. 8a). The AG enriched codons do not show noticeable bias though (Supplementary Fig. 8b). However, despite the fact that polypurine sequences are less enriched in the first 75 codons, the ribosomes are still more likely to be stalled, in response to RocA, at the first 75 codons (Supplementary Fig. 8c, d). As we are proposing in this study, clamping of eIF4A onto the poly-purine sequences in the 5′ CDS regions induces ribosome stalling during early elongation, which should reduce the number of the ribosomes moving further down to the 3′ CDS regions. In other words, repression of translation elongation naturally reduces the ribosome footprints in the downstream CDS regions. By contrast, ribosome accumulation could still take place in the upstream regions of the RocA-induced eIF4A binding events.

HEK293 cells with eIF4A1 double mutations (Phe163Leu-Ile199-Met) were shown to be resistant to RocA treatment[41]. Interestingly, our metagene analysis with the ribosome profiling of eIF4A1 double mutant cells[41] showed that the biased accumulations of ribosomes in the 5′ UTR and CDS regions in response to RocA treatment were largely abolished (Supplementary Fig. 2b, Supplementary Fig. 9a). More specifically, ribosome stalling in the 5′ UTR of IRGs and CDS of ERGs were both relieved (Fig. 4e). For example, ribosomes stalled in the 5′ UTRs of the IRGs PARP1 or ENO1 and in the CDSs of ERGs UQCC2 or NDUFS6 were completely or partially resolved (Supplementary Fig. 9b, c). In addition, the events of ribosome stalling induced by RocA in the upstream region of the poly-purine motifs in the 5′ UTR of IRGs and CDSs of ERGs were almost completely lost under the context of eIF4A mutation (Fig. 4f, Supplementary Fig. 10a, b). Taken together, the

results above indicate that ribosome stalling at the elongation stage induced by RocA should also be associated with direct binding of eIF4A to the poly-purine sequences in the CDS.

As shown by Fig. 3, in general, the ERGs and the IRGs exhibit differential codon usages, charges, and hydrophobicity, suggesting that compared to the IRGs, the ERGs are subjected to suboptimal translation elongation. Here we further showed that the ribosome stalling events upon RocA treatment take place mainly in the upstream of poly-purine sequences of the ERGs, an effect being lost under the context eIF4A mutation. This strongly indicates that the poly-purine sequences are the primary factors that drive the elongation inhibitory effect of RocA in a manner depending on eIF4A.

To further confirm this conclusion, we analyzed the sequence features of the regions enriched by ribosome footprints (RocA vs DMSO ≥ 1.5 fold) within the first 50 codons of the ERGs upon 0.3 μM RocA treatment. The regions with depleted ribosome footprints (RocA vs DMSO ≤ 0.67 fold) serve as references for comparison. No significant difference of the local cAI, tAI, charge, and hydrophobicity were observed between these two sets of regions in CDS of the ERGs (Supplementary Fig. 11). Therefore, the features of translation elongation, such as codon usage, charge, and hydrophobicity, do not seem to play determining roles in mediating the inhibitory effects of RocA on translation elongation. They rather provide a suboptimal overall context for slow translation elongation, in which ribosome stalling due to eIF4A binding to the poly-purine motifs in response to RocA is more likely to be observed.

## Perturbed translation elongation of the ERGs by RocA led to reduced protein synthesis

The results above illustrate that in addition to the canonical function of RocA in blocking translation initiation, disruption of translation elongation is another major mode of action of RocA. Specifically, the unbalanced accumulation of ribosome during early translation elongation in response to RocA indicates ribosome stalling, which should lead to reduced rates of protein synthesis[4,54]. To confirm the inhibition of ERG protein synthesis due to disrupted translation elongation, we performed a set of protein synthesis reporter assays. Specifically, ribosome stalling signals such as poly-AG motifs from several ERGs were cloned into firefly luciferase cDNA downstream of the start codon (Fig. 5a). The reporters do not have any G-quadruplex, which might interfere with translation elongation, in the first 50–60 codons of the CDSs containing the poly-purine motifs. Note that the HCV-like IRES, which could initiate translation by binding to the solvent side of 40S ribosomes, was used for translation initiation of the firefly and Renilla luciferase reporters. Such IRES-dependent and cap-independent translation initiation is independent of translation initiation factor eIF4A1[55] and shown to be unaffected by RocA[40]. 12 h after transfections of these reporter plasmids, RocA treatment for 30 min resulted in significantly reduced firefly luciferase levels, in relative to the Renilla luciferase levels as references (Fig. 5b). For the purpose of comparison, RocA treatment induced similar efficacy of translation repression for the luciferase reporters with polypurine motifs found in 5′ UTRs of 3 IRGs (Fig. 5b), which is consistent to previous reports of RocA-induced translation initiation repression[40]. Furthermore, replacement of the poly-AG motifs in the CDS by poly-CT (Fig. 5c) resulted in potent

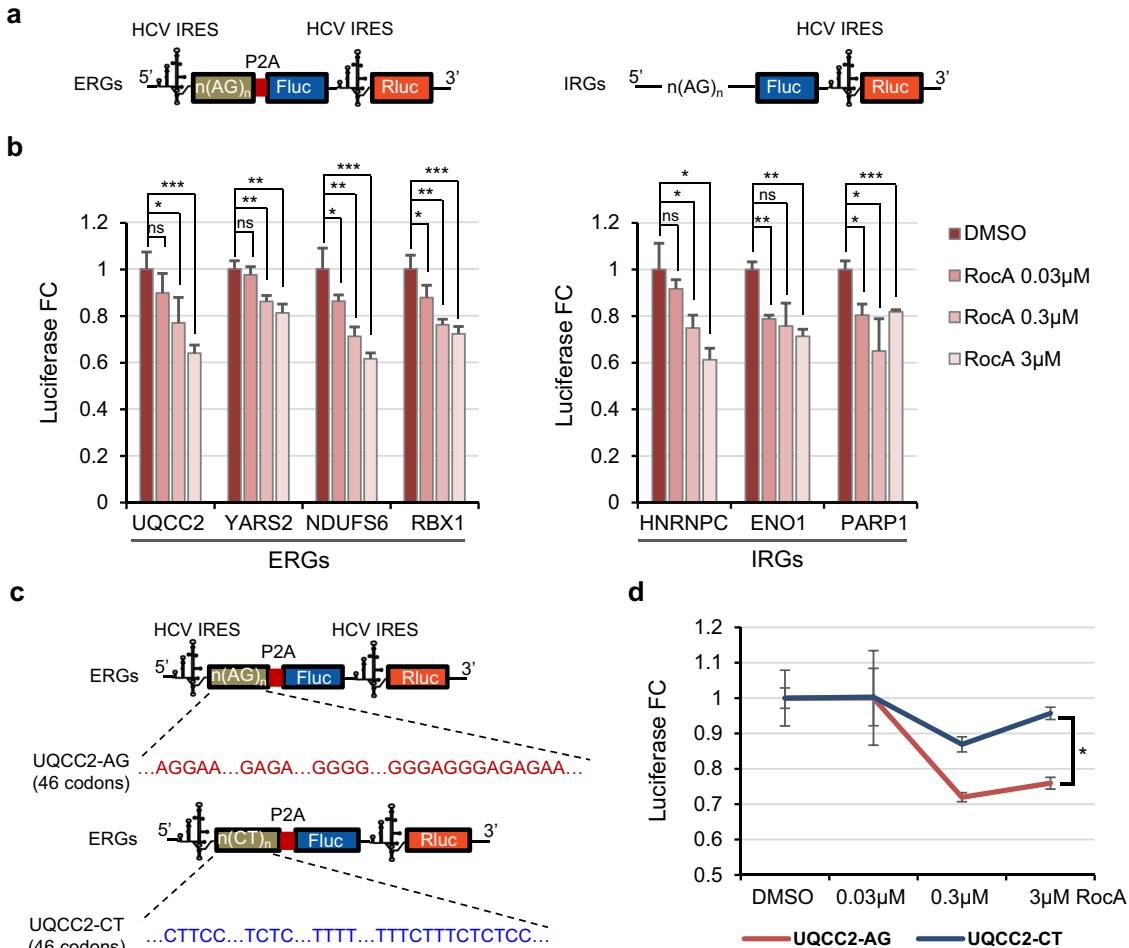

**Fig. 5 | Luciferase reporter assays showing reduced protein synthesis due to blockage of translation elongation upon RocA treatment. a** Structures of dual-luciferase reporters with poly-purine sequences obtained from upstream CDSs of ERGs or 5′ UTRs of IRGs. HCV-like IRES was used for cap-independent translation initiation. **b** Firefly luciferase activities in relative to Renilla luciferase after transfections of different constructs, followed by 30 min of RocA treatments at different doses. **c** Design of the reporter constructs for rescue experiments by replacing the poly-purine sequences with poly-pyrimidine in the CDS regions. **d** Firefly luciferase activities with the constructs in panel c in response to different doses of RocA. Data are presented as mean values ± SD. "ns", no significance; *$p \leq 0.05$; **$p \leq 0.01$; ***$p \leq 0.001$. *P*-values are all calculated by two-sided Student's *t* test without adjustments. $n = 3$ biologically independent samples. Source data are provided as a Source Data file.

resistance of the firefly luciferase levels to RocA treatments (Fig. 5d). This confirms that the reduced protein synthesis upon RocA treatment is dependent on the poly-AG motifs in the 5′ CDS of the ERGs.

Taken together, the results above illustrate that by shifting the ribosome occupancy patterns along the mRNA transcripts during translation initiation and early elongation, the gene-specific dual functions of RocA should both result in a reduction in the protein synthesis rates. It is worth noting that given the long half-life of proteins (median half-life about 7 h), translation inhibition by RocA for such a short period (30 min) would not result in reduction of protein abundance that is detectable by regular mass-spectrometry or western blotting, especially for the proteins with high abundance such as ribosome proteins and mitochondrial proteins. On the other hand, prolonged treatment of RocA would result in more secondary effects, including transcriptional repression of genes[27], which would make it difficult, if not impossible, to evaluate the change of protein abundance due to repression of translation.

### Suppression of translation elongation induced by RocA is preserved in cancer cells

RocA has been reported to have potent anticancer activities in many tumors[26–31,36,56], which were mainly attributed to the inhibition of translation initiation[24]. Based on previously published ribosome profiling data from two lung cancer cell lines (NCI-H1650, NCI-H520) and two melanoma cell lines (A375, Hs936T)[42], we confirmed that RocA induces global shifts of the ribosome distribution patterns in these cancer cell lines (Supplementary Fig. 12a, b), which is consistent with our observations discussed above.

Similar metagene analyses with the ERGs and IRGs identified in HEK293 cells further confirmed the gene-specific dual functions of RocA on translation initiation and early elongation in cancer cells (Fig. 6a). In addition, the increased ribosome densities upstream of poly-purine motifs in the CDS of ERGs or 5′ UTR of IRGs were also reproduced in the four cancer cell lines upon RocA treatment (Fig. 6b), suggesting similar eIF4A-dependent machinery of RocA-induced translation perturbation. Thus, the results above indicate the same dual mode of action of RocA in the noncancerous HEK293 cells and in cancer cell lines. Given that ERGs are enriched by essential processes such as ATP synthase and the mitochondrial respiratory chain complex, blockage of ERGs translation elongation should also lead to suppression of fundamental cellular processes such as energy production, metabolism, and proliferation. Therefore, we propose that the identified dual mode of action of RocA should account for its antitumor effect.

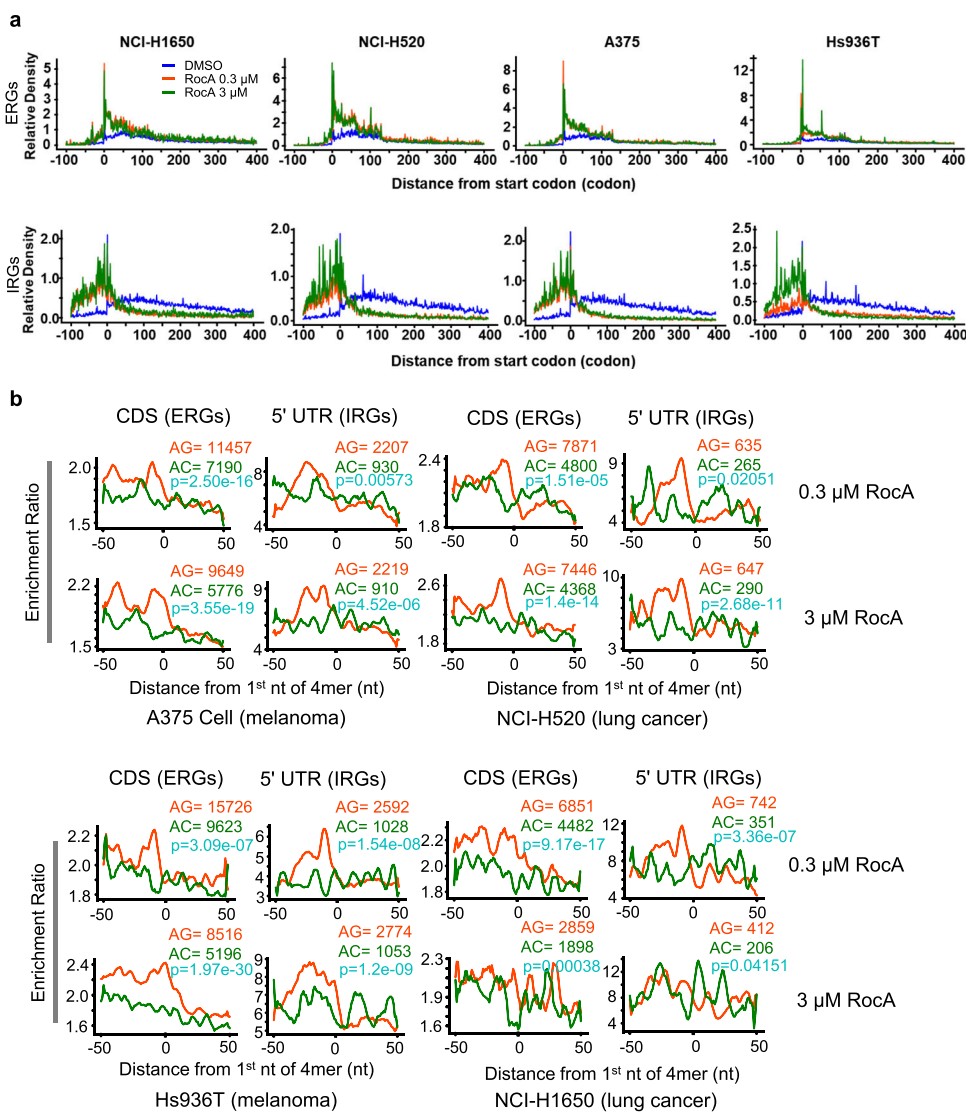

**Fig. 6 | Perturbation of translation elongation induced by RocA in lung cancer and melanoma cells. a** Metagene plots of ERGs and IRGs with ribosome profiling data from different cancer cell lines. Melanoma: A375 cells and Hs936 cells; lung cancer: NCI-H1650 cells and NCI-H520 cells. **b** Enrichment ratio of ribosome density around poly-purine motifs (4mer) by comparisons between RocA and DMSO in A375, Hs936T, NCI-H520 and NCI-H1650 cells. *P*-values are all calculated by two-sided Student's t test without adjustments. Source data are provided as a Source Data file.

## Discussion

Our understanding of the translatome and its complicated regulation has quickly increased for the past decade, benefitting greatly from developed high-throughput techniques such as ribosome profiling. Quantitative profiles of the translatomes at the sub-codon resolution have accumulated from these cutting-edge studies of translation regulation in a large variety of biological contexts. The experimental settings of these studies have covered various types of translation regulation taking place at different stages, including ribosome complex assembly, ribosome scanning, translation initiation, ribosome stalling, termination, etc. Therefore, the collection of previous data now provides an unprecedented resource for mining the representative patterns for different types of translation regulation. Such data collection can serve as a reference for elucidating the mode of translation regulation under a particular experimental or physiological condition. For example, the present study revisited the ribosome profiling data of RocA treatment, not by looking into the specific condition itself but by evaluating its position among many other conditions of translation perturbation. The results clearly demonstrated a potent function of translation elongation stalling by RocA, similar to other conditions that have been well characterized to inhibit translation elongation (Fig. 1).

As a class of natural compounds, RocA and its derivatives have been well characterized by their potent antitumor activities in multiple hematological cancers and solid tumors[26–32,34–37,56,57]. Global inhibition of protein synthesis has been found to underlie the antitumor effect of RocA[24]. Recent studies further demonstrated the machinery of RocA in repressing translation initiation by clamping eIF4A on the poly-purine motifs in the 5′ UTRs of mRNA transcripts[40,41,43]. Here, we further uncovered a mode of action of RocA, which is executed in the translation elongation of a different group of transcripts (ERGs) bearing a series of features, such as short 5′ UTRs, suboptimal codons, and slow elongation rates. Functionally, these genes are involved in highly essential processes, such as those associated with mitochondrial activities, which we believe is partly accountable for the potent antitumor effect of RocA.

Previous studies have well demonstrated that RocA clamps eIF4A to the poly-purine sequences in the 5′ UTR, thereby blocking the scanning of 43S PIC and translation initiation as characterized by abnormal accumulation of 80S ribosomes in 5′ UTRs due to enhanced

uORF translation[40,41]. However, this model did not explain why translation of the mRNAs without poly-purine motifs in their 5' UTRs was also inhibited by RocA. Interestingly, our metagene analyses with previous data from eIF4A1 double mutant (Phe163Leu-Ile199Met) cells indicated that similar to the effect of translation initiation inhibition, the stalling of translation elongation induced by RocA is also dependent on its binding partner eIF4A (Fig. 4e, f and S9, S10). Furthermore, integrative analyses of the previous ribosome profiling and iCLIP-seq datasets confirmed that the events of ribosome stalling in the 5' UTR and the CDS regions were both associated with the direct binding of eIF4A to the poly-AG motifs. Therefore, we propose a mode of translation inhibition functioning at the stage of translation elongation, which completes the machinery of RocA inducing global translation repression via eIF4A.

Previous studies also proposed an alternative mechanism, i.e., RocA traps the eIF4F complex at the 5' cap and inhibits the recruitment of 43S PIC, which leads to sequestration of eIF4A and thereby global translation suppression due to limited supply of free eIF4A (namely, a "bystander effect")[43]. However, for the mRNAs whose translation initiation was eIF4A- and scanning independent, e.g., mRNAs with TISU elements, their translation inhibition upon RocA treatments was still not well explained. Instead, although the potential "bystander effect" was not precluded by our analyses, the dual modes of action of RocA proposed in the present study were better supported by the observed patterns of ribosome occupancy on both the IRGs and ERGs (Fig. 1e, f, Supplementary Fig. 1 and Supplementary Fig. 2d). In particular, for the mRNAs with TISU elements, their translation elongation was blocked by RocA due to the binding of eIF4A to the poly-purine motifs in the CDS, whereas their translation initiation was eIF4A- and scanning independent. Recently, it has been shown that eIF4A could promote bidirectional PIC scanning for start codon selection[58], which serves as an operational mechanism for translation initiation of the mRNAs with ultra-short 5' UTR. However, it is also well demonstrated that the mRNAs with short 5' UTR tend to have very low translation efficiency unless they bear TISU[46]. Such TISU-dependent translation fits in the cap-dependent slot-in model of translation initiation with minimal scanning. Therefore, based on these prior studies, although we cannot completely rule out the backward scanning, it is disputable whether such eIF4A-dependent bi-directional scanning is a dominant factor in determining the translations of the mRNAs with TISU elements. Nevertheless, our study is focused on the ribosome occupation patterns during translation elongation perturbed by RocA, whereas the uni- or bi-directional scanning affects translation initiation.

The present study demonstrated the function of RocA in inducing ribosome stalling on the CDS regions and furthermore, accumulation of disomes. Ribosome stalling is known to trigger the RQC pathway, in which stalled ribosomes are recognized by ZNF598 (Hel2 in yeast) and split by the ASC-1 helicase complex (Slh1/Rqt2 in yeast) or the recycling factors Pelota, HBS1L (or GTPBP2) and ABCE1 (Dom34, Hbs1 and Rli1 in yeast) into 60S and 40S subunits, eventually leading to termination of translation. Interestingly, many of the genes involved in RQC were subjected to translation perturbations by RocA (Supplementary Fig. 13a), especially those responsible for ribosome splitting, such as ASCC3, PELO, HBS1L, ABCE1, and ASCC2[59–64]. Some genes, such as ASCC3, were inhibited by RocA at translation initiation by the canonical model proposed previously[40,41], as illustrated by the accumulation of ribosomes in the 5' UTRs and a decrease in ribosome densities in the CDS upon RocA treatment (Supplementary Fig. 13b). On the other hand, other RQC genes, such as EDF1, showed patterns of ribosome stalling in their CDS, indicating perturbed translation elongation (Supplementary Fig. 13c), leading to increases in their total ribosome densities (Supplementary Fig. 13a). Therefore, taken together, the results above suggest that the first response of the translation dynamics triggered by a 30-min treatment of RocA should lead to an impaired RQC pathway in longer terms, which in turn might result in failure of the cells to efficiently resolve the collided ribosomes and alleviate the stress of ribosome stalling, thereby strengthening the toxicity of RocA.

Potent antitumor effects of RocA have been frequently reported before, especially in hematological cancer cells, whereas non-cancerous cells showed strong resistance to RocA[24,26,28,33,34]. The mechanism accountable for such differential responses of malignant and noncancerous cells remains unclear. This may be simply attributed to the expression levels of RocA targets (eIF4A1, eIF4A2 and DDX3X), which are much higher in hematological cancer tissues than in normal tissues (data not shown). Interestingly, although the mechanism remains elusive, it has been shown that RocA preferentially activates p38 and JNK in malignant but not in normal primary cells, thereby inducing cell apoptosis[24,26]. Recent studies found that upon ribosome stalling under stress conditions, the disomes formed by collided ribosomes serve as a platform to recruit ZAKα, a MAPKKK family protein that activates p38 and JNK, leading to cell apoptosis[9,65]. The present study demonstrated the role of RocA in inducing ribosome stalling on the CDS regions in HEK293 cells and a series of other cancerous cells, leading to the accumulation of disomes. We propose that these disomes induced by RocA should be accountable for the activation of p38 and JNK by recruiting ZAKα. Interestingly, a recent study confirmed that the phosphorylation of p38 induced by un-resolved colliding ribosomes was completely abolished after knockdown of ZAK[66]. Therefore, based on the theory above, the expression levels of ZAKα may be critical in determining whether the disomes induced by RocA are cytotoxic. Further biochemical studies are needed to validate this hypothesis for the highly specific antitumor effect of RocA.

## Methods

### Library preparation for Disome-seq

HEK293FT cell lines were cultured in DMEM with 10% fetal bovine serum in 5% $CO_2$ incubator at 37 °C. Cells were treated with 0.3 µM RocA (MCE) for 30 min when the confluence was at 70-80%. After the treatment, cycloheximide at 100 µg/mL was added into the medium and incubated for 5 min. Cells were then rinsed twice by cold PBS containing 100 mg/mL cycloheximide. Finally, the cells were collected by scraping and centrifugation at $300 \times g$ for 5 min at 4 °C.

Cell pellets were lysed with lysis buffer (20 mM Tris, 150 mM NaCl, 5 mM MgC12, 1 mM DTT, 25 U/mL DNaseI, 100 µg/mL cycloheximide, 1% (V/V) TritonX-100, 0.1% (V/V) NP-40) for 10 min on ice, followed by centrifugation at $13,000 \times g$ for 10 min at 4 °C. Five $OD_{260}$ units of the supernatant was treated with 450 units of RNaseI (Ambion:AM2295) for 45 min at room temperate, and the reaction was stopped by SUPERase•In (Ambion: AM2694). The ribosome-RNA complex was isolated and enriched by Sephacryl S400 columns (GE Healthcare), followed by RNA extraction with Trizol (Invitrogen: 15596018). The RNA was then resolved on 15% urea polyacrylamide gel, and the fragments with ~58–70 nucleotides were extracted with gel extraction buffer (400 µL Nuclease-Free Water, 40 µl of 5 M Ammonium Acetate and ~2 µl of 10% SDS) by incubation with agitation at 37 °C overnight. Finally, the RNA was precipitated with 700 µL isopropanol and 2ug glycogen. Sequencing libraries were then prepared as described in previous reports[19].

### Luciferase reporter assays

To construct luciferase plasmids for translation reporters, CDS sequences of ERGs near the start codons containing poly-purine (poly-AG) were inserted between SV40 promoter and ORF of Firefly luciferase in pEZX-MT06 (GeneCopoeia). HCV-IRES was inserted between the SV40 promoter and CDS sequences for cap-independent translation. For rescue experiments, the poly-purine sequences in the 5' CDS of the ERG UQCC2 were replaced with poly-pyrimidine sequences. All the inserted sequences mentioned above are supplied in Supplementary Data 4.

HEK293FT cells cultured in 96-well plates were subjected to forward transfections of the luciferase reporter plasmids with Lipofectamine 2000 (Invitrogen) according to the manufacturer's instruction. Sixteen hours after the transfection, the cells were treated with RocA at three concentrations, 0.03 μM, 0.3 μM, and 3 μM. Thirty minutes after RocA treatment, cells were washed with PBS and lysed with lysis buffer (VazymE) for 15 min at room temperature. The luciferase assay was performed with Dual-Luciferase Reporter Assay System (VazymE) according to the manufacturer's instructions. Luminescence was detected with SPARK System (TECAN). HCV-IRES Renilla were used as internal controls.

## MS2bp-GFP RNA pull down

MS2-GFP-RIP assays with the CDS regions of 3 ERGs fused with MSbs sequences were performed as previously described[67]. In details, the empty plasmid pcDNA3.1-MS2 or the plasmid with the selected CDS regions inserted was co-transfected with pMS2-GFP (Addgene) into HEK293FT cells. Forty-eight hours after transfection, cells (10 million) were harvested and lysed with 1 ml native lysis buffer (50 mM Tris pH7.4, 150 mM NaCl, 0.5% NP-40, 0.5 mM PMSF, 2 mM RVC, protease inhibitor cocktail (Roche)) followed by sonication. The lysate was centrifuged at $10,000 \times g$ for 30 min at 4 °C, and the supernatant was pre-cleared with 10ul Dynabeads Protein-G. The samples were then incubated with GFP antibody (3 μg per reaction; ab290, abcam) for 2 h at 4 °C, followed by addition of 20 μl Dynabeads Protein-G to the mixture and incubation overnight at 4 °C on a rotating shaker. Next, the samples were washed with wash buffer (50 mM Tris pH7.4, 300 mM NaCl, 0.5% NP-40, 0.5 mM PMSF, 2 mM RVC, protease inhibitor cocktail (Roche, 4693124001)) for 4 times at 4 °C. Finally, the magnetic beads were resuspended in 20 μl 1× SDS loading buffer and boiled for 10 min. The samples were analyzed by 10% SDS-PAGE and probed by immunoblotting with anti-eIF4A1 (1:1000, orb353605, Biorbyt).

## Collection of ribosome profiling datasets

Twenty-four ribosome profiling datasets containing 49 conditions of translation perturbation were downloaded from Gene Expression Omnibus (GEO), and the detailed information of these datasets is provided in Supplementary Data 1. Specifically, all inhibitors of mTORC1 signal pathway are labeled as "Initiation" conditions, given that mTORC1 mainly regulate translation initiation through phosphorylation of 4EBP1 and p70S6K1/2[50,68–71]. Amino acid starvation can also inhibit translation initiation via both mTORC1 signal pathway[70,72] and GCN2-eIF2α pathway[73,74], and therefore it is also labeled as "Initiation". Other conditions known to inhibit translation elongation were labeled as "Elongation". In total, there are 29 conditions of translation initiation, 15 of translation elongation, and 5 conditions of RocA treatments, including 3 different concentrations (RocA003: 0.03 μM, RocA03: 0.3 μM, RocA3: 3 μM) in HEK293 cells and RocA treatment in Aglaiacized HEK293 cells with double mutations of eIF4A1.

## Pre-processing of the ribosome and disome profiling data

All the ribosome profiling data were processed with the same pipeline[75,76]. The human reference genome (hg38), non-coding sequences (ncRNA) and the annotation file are downloaded from the Ensembl Genome Browser (release 88, https://www.ensembl.org/index.html)[77]. FastQC (http://www.bioinformatics.babraham.ac.uk/projects/fastqc/) was used for quality control and the adapter sequences were trimmed using the cutadapt program[78]. Reads with length 15–45 nt were used, but only those with 3-nt periodicity were counted, which was assessed by RiboCode[79,80]. For disome-seq, the reads with length 30–70 nt were kept for the following analyses. Low-quality reads with Phred quality scores lower than 25 (>75% of bases) were removed using the fastx quality filter (http://hannonlab.cshl.edu/fastx_toolkit/). The

sequence reads originating from rRNAs were identified and discarded by aligning the reads to human ncRNAs using Bowtie 1.1.2[81] with no mismatch allowed. The remaining reads were mapped to the genome and transcriptome using STAR[82] with the following parameters:– runThreadN 8–outFilterType Normal–outWigType wiggle–outWig Strand Stranded–outWigNorm RPM–alignEndsType EndToEnd–out FilterMismatchNmax 1–outFilterMultimapNmax 1–outSAMtype BAM SortedByCoordinate–quantMode TranscriptomeSAM GeneCounts–out SAMattributes All. Then, the BAM format files were sorted and indexed using Samtools (http://www.htslib.org/).

The P-sites of ribosome footprint (RPF) were determined by RiboCode[79,80]. The counts at each codon, coverage of each transcript, and total RPF counts of each gene excluding the first 15 codons and the last 5 codons were obtained with scripts from RiboMiner[76]. Differential expression analyses were performed by DESeq2[83], with the cutoff of | log2FC| ≥ 1 and $q$ value ≤ 0.05. RPKM values of each gene were computed by custom python scripts.

## Metagene analyses of ribosome profiling data

Metagene analyses of ribosome profiling data were performed by *MetageneAnalysis* of RiboMiner[76] with "-u 100 -d 400 -l 100 -n 10 -m 1" parameters. Specifically, for genes that exhibited more than one transcript, only the longest transcript was used in all analyses. Only the transcripts with length over 100 codons and RPKM more than 10 in the whole CDS region were considered. The ribosome densities were normalized to reads per million mapped reads (RPM).

## Polarity calculation

To quantify patterns of ribosome occupancy along transcripts, we computed the polarity score for every gene as described before[11]. Polarity scores of transcripts for all the ribosome profiling datasets were calculated by RiboMiner[76]. As for each transcript, the change of polarity score (Δpolarity) upon a particular treatment condition was calculated by comparing the polarity scores in the treatment group and the control group.

## Hierarchical clustering of the translation perturbations

Hierarchical clustering of the 49 conditions of translation perturbation was performed based on the polarity score differences (Δpolarity) of the genes. Clustermap of seaborn package in python was used with the 'ward' method and 'euclidean' metric. Transcripts with RPKM more than 1 in all 49 condition pairs were selected first. Next, only the transcripts (4842 in total) with mean coverage among all condition pairs larger than the mean coverage of all transcripts in all condition pairs were used for the clustering analysis.

## Calculations of ribosome and disome occupancy patterns around poly-purine motifs

Ribosome or disome density around poly-purine motifs was calculated using custom python scripts. We first searched for the short poly-purine motifs in the CDSs and 5′ UTR of both ERGs and IRGs, and the ribosome densities within the up- or down-stream 50 nt from the first nucleotide of the poly-purine motifs were calculated. Only the poly-purine motifs with at least 10 RPF counts falling in this 101-nt region were used for further analyses. The first and last 50 nt of the CDSs and 5′ UTRs were excluded. For disomes, the up- or down-stream 100 nt from the first nucleotide of the poly-purine motifs were calculated. Enrichment ratio of the ribosome or disome density around poly-purine motifs was calculated by comparing the treatment to the control groups. Statistical significance was also assessed using the t-test assuming two-tailed distribution and unequal variance.

## Processing of iCLIP-seq data

The iCLIP-seq data for eIF4A was downloaded from GEO (GSE79392). Common practice of CLIP-seq data has been followed[84]. The low-

quality reads and rRNA reads from iCLIP-seq data were discarded first. The remaining iCLIP-seq reads were then aligned to both CDS of ERGs and 5' UTR of IRGs using BLAST[85] with no mismatch allowed, respectively. We then looked for the over-represented short poly-purine motifs (4–6 nt) in the iCLIP-seq reads, and the motif enrichment was calculated by the ratio of motif frequency between libraries as described before[40]. Statistical significance was assessed using the t-test assuming two-tailed distribution and unequal variance.

## Characterization of sequence features

The local tRNA adaptation index (tAI), codon adaptation index (cAI), and the average charge and hydrophobicity of amino acids were all calculated by RiboMiner as described before[76]. The R package ggpubr[86] was used for the plots. Minimum free energies were calculated with RNAfold (ViennaRNA Package[87]). The statistical significance levels were assessed using the wilcoxon rank sum test in R platform. The ratios of ribosome density between RocA and DMSO samples at each position of ERGs were calculated by RiboMiner[76], after which these positions with a ratio ≥1.5 at the first 50 codons of CDSs of ERGs were defined as "Ribosome-enriched-positions" and positions with a ratio ≤0.67 were defined as "Ribosome-depleted-positions". Sequence features such tAI, cAI, charge and hydrophobicity of "Ribosome-enriched-positions" and "Ribosome-depleted-positions" were analyzed.

## Sequence motif enrichment analysis

Sequence motifs enriched in 5' UTR and CDS regions were assessed by HOMER[88]. Sequence logos of nucleotides were regenerated by R package seqLogo[89], and the amino acid sequence logos generated by Seq2Logo.

## Gene ontology enrichment analysis

Gene ontology (GO) enrichment analyses were performed either by clusterProfiler[90] or Metascape[91]. Multiple testing corrections were performed with the Benjamini–Hochberg method. The plots were regenerated by Microsoft Excel.

## Reporting summary

Further information on research design is available in the Nature Portfolio Reporting Summary linked to this article.

## Data availability

Information of all the public ribosome profiling and iCLIP-seq datasets used in the present study is provided in Supplementary Data 1. The human reference genome (hg38), non-coding sequences (ncRNA) and the annotation file are downloaded from the Ensembl Genome Browser (release 88, https://www.ensembl.org/index.html). The Disome-seq data generated in this study has been deposited into the GEO database under accession ID GSE201364. Source data are provided with this paper.

## Code availability

Codes for all the data analyses in this study are available in github (https://github.com/xryanglab/RocA).

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

## Acknowledgements
We would like to thank professor Nicholas Ingolia at UC Berkeley for discussions about the mechanisms of translation regulation by RocA and the processing of iCLIP-seq data. We thank the supports from the Tsinghua University Branch of China National Center for Protein Sciences (Beijing) and Tsinghua University Technology Center for Protein Research, including the core facilities of Biocomputing, Genome Sequencing and Analysis at Tsinghua University. This work was funded by the National key research and development program, Precision Medicine Project (2016YFC0906001), the Tsinghua University Spring Breeze Fund, the Tsinghua University Initiative Scientific Research Program (2019Z06QCX01), and the National Natural Science Foundation of China (81972912 and 31671381).

## Author contributions
Conceptualization, F.L. and X.Y.; Bioinformatics analysis, F.L. and S.H.; Experimental validation, J.F., Y.Y., Q.Zo, Q.Ze; Writing-original draft, F.L. and X.Y.; Writing-review, modification, and editing, F.L., J.F., Y.Y., X.Y.; Visualization, F.L and X.Y.; Supervision, X.Y.; Funding acquisition, X.Y.

## Competing interests
The authors declare no competing interests.
