## [Peer Review File · Nature Communications]

Reanalysis of ribosome profiling datasets reveals a novel function of rocaglamide A in perturbing the dynamics of translation elongation via eIF4AREVIEWER COMMENTS

Reviewer #1 (Remarks to the Author):

Rocaglates have attracted great interest due to their potential for tumor treatment and anti-COVID19 effects. Re-analyzing published data from Iwasaki and Ingolia's group with the novel disome profiling data of their own, Li et al. elegantly showed that rocaglates may inhibit the translation elongation. The mechanistic detail should be a splendid addition as a novel mode of translation repression by the leading compounds. This reviewer recommended addressing the following issues before the publication.

Major point:

1. This reviewer is deeply concerned about the authors' interpretation of ribosome footprints in 5' UTR. In the earlier paper (Iwasaki et al. Nature 2016, Wolfe et al. Nature 2014), those reads were attributed to "80S" ribosomes on upstream ORF (uORF), but not scanning 40S ribosomes, since 40S footprints were hard to recover without crosslinking as shown in TCP-Seq (Archer et al. Nature 2016 and other many manuscripts). Since authors referred to the footprints in 5' UTR as 40S footprints, the authors should rewrite the manuscript all through the manuscript.

2. Related to the point above, the enhanced uORF translation by rocaglates has been suggested as an additional parameter to promote translational repression at the initiation step, but not the primary reason (Iwasaki et al. Nature 2016). Clamping of eIF4A to polypurine motif (this effect per se could not be monitored by ribosome profiling) should be what happens first (then this may lead to uORF translation but not always). Thus, 80S reads on 5' UTR should be a direct indicator of rocaglate sensitivity in translation initiation.

3. The nomenclature of "RPF upregulated" gene (and also "RPF downregulated" gene) should be carefully used. Rocaglates (or other translation inhibitors too) definitely leads to global translation repression (even for the RUGs). Deep sequencing is always "relative" unless spike-in controls are added to library preparation. Thus, the correct wording here should be less sensitive/high sensitive or equivalent.

4. Regarding the short 5' UTR genes, the authors classified them as TISU which does not require eIF4A and scanning. However, currently, this model is challenged by the report of Gu et al. Nat Commun 2021 that showed the backward (3'-to-5') scanning. The authors carefully characterized those mRNAs, scanning independency, and the interpretation of the data.

5. Figs 3 and 4 indicated the important parameters (charge, hydrophobicity, and polypurine motif) that are associated with ribosome stalling by rocaglates. The authors should clarify which factor is the primary one and which is additive to enhance the primary effect. Detailed data analysis and/or reporter assay may be helpful to dissect the role of these parameters.

6. Although the authors nicely showed that eIF4A clamping on CDS by rocaglates by re-analysis of iCLIP data, biochemical validation should be helpful. This could be conducted by toe printing assay of eIF4A with in vitro translation system or equivalent.

7. Given that polypurine in 5' UTR could evoke translation repression at the initiation step too, the authors should evaluate the efficacy of translation repression at the initiation step and elongation step and address which one is more dominant. This could be addressed by introducing synthetic polypurine motifs in 5' UTR or CDS in reporters, comparing the potency in repression by rocaglates.

Minor point:

1. Although authors mentioned about the ribosome collision and the subsequent RSR/RQC pathway

activation in the discussion section, they noted that this may not be led because of the high sensitivity of related components in translation. This was a contradiction.

2. Related to the point above, in addition to ribosome stalling, rocaglates should suppress ribosome load by translation initiation inhibition. This should reduce the probability of ribosome collision (which is naturally taken by eIF2alpha phosphorylation by RSR and eIF4E2-mediated repression in RQC). Authors should provide experimental evidence that rocaglate leads to RSR/RQC. Otherwise, authors should avoid the discussion regarding RSR/RQC.

3. For Fig. 4E, blue line (for Aglaia-cized-DMSO) was hard to see. Please consider better coloring of the graph.

4. Please double-check the following figure citation in the manuscript.

Fig. 5E in line 20 at p12 should be Fig. 4E;

Fig. 5A in line 4 at p15 should be Fig. 5C;

Fig. 5B in line 5 at p15 should be Fig. 5D

5. For Fig. 5, please highlight which IRES was used for the experiments and the mechanism of the IRES in the main text.

6. Regarding Fig. 4C (and S6), authors should consider showing the same plots for CDS for RDGs and 5' UTR for RUGs, for control.

7. Regarding Fig. 4E, the authors should consider drawing the similar meta-analysis around as shown in Fig. 4C and 4D.

8. To confirm that RUGs are translationally repressed even though their footprint change looks like translation upregulation, western blotting of the endogenous targets of rocaglates would be appreciated.

9. This reviewer recommends the authors examine the positional effect of the polypurine motif in CDS. More pronounced results may be obtained in motif enrichment (Fig. 4A-B) and footprint accumulation around polypurine motif (Fig. C-D) when focusing on CDS near from start, compared to the remaining parts.

Reviewer #2 (Remarks to the Author):

Li, Fang, and Yu et al identify a new mode of translational inhibition by Rocaglamide A (RocA) in this work. Prior work found that RocA inhibits translation by binding to eIF4A and clamping it onto polypurine motifs in 5' UTRs. This work identifies a second mode of translational inhibition by RocA, which the authors propose is mediated by eIF4A-dependent interactions with polypurine motifs in early coding sequences. This finding is interesting and in general the experiments and analyses are performed well and this work should be published. Comments below seek to improve the work, with the comment about the mechanism the most important to address in my opinion.

Major comments:

If RocA is generally inducing a block in elongation by eIF4A clamping to polypurine motifs it is confusing why the accumulation of ribosomes is towards the start of the CDS rather than throughout the entire CDS. It would seem an elongating ribosome could be stalled by a clamped eIF4A anywhere in the CDS. An alternative explanation might be that stalls in scanning by eIF4A clamping promote initiation at upstream out of frame start codons, leading to accumulation of ribosomes early in the

CDS until they encounter an out of frame stop codon. Prior work showed RocA causes changes to uORF initiation. Can the authors explore this possibility by analyzing out of frame start codons or the frame of ribosome protected footprints in RUGs or otherwise? Why else might ribosomes selectively accumulate near the start of the CDS? Is there a 5' bias in the distribution of polypurine motifs within CDSs? What would happen if the polypurine reporters were constructed with the polypurine in the middle or end of the CDS? Fully understanding this may be beyond the scope of this work and that's ok, but additional exploration seems warranted to better define the mechanism as distinct from an initiation defect.

It would be helpful in the reporter experiments to also include canonical RocA 5' UTR clamping reporters. This would allow a side-by-side comparison of the magnitude of effect on 5' UTR clamping versus the new mode of RocA action identified in this work.

The authors note changes in regulation to RQC-related genes, which is interesting. A component of the RQC involves mRNA degradation of target mRNAs. If the authors have RNA-seq data from these experiments, they could use a computational package such as INSPECT (PMID 25957348) to estimate changes in RNA stability. It would be interesting either way – perhaps RocA is inducing mRNA decay in RUGs, or perhaps the changes to RQC-related genes are blocking mRNA decay. If the authors do not have RNA-seq data, generating it is beyond the scope of this work in my opinion. If the authors wished, they could measure select transcript half-lives with EdU labeling or transcriptional shut-off, though this is also unnecessary.

Minor comments:

It would increase the readability of the figures if the legends of RocA3, RocA03, and RocA003 were changed to 3 μ M RocA, 0.3 μ M RocA, and 0.03 μ M RocA throughout the paper.

The reporter experimental format leaves open the possibility that the actual effect may be larger than is shown in Fig 5. Specifically, cells were transfected for 12 hours and RocA was only applied for 30 minutes. During this experiment, luciferase will accumulate prior to addition of RocA and translation will only change for 30 minutes. Is it possible to treat the cells for a longer duration with RocA? This may increase the relative impact of RocA on translation.

Reviewer #3 (Remarks to the Author):

In the manuscript by Li et al, the authors reanalyzed a large number of previously published Riboprofiling data, and found some new insight on the regulation of translation elongation. Specifically, they found that treatment of Rocaglamide A (RocA), a natural product that selectively inhibit translation initiation by clamping eIF4A onto the poly-purine motifs of the 5'UTRs, can also increase the eIF4A occupancy on the coding region. They further found that such stall of eIF4A on ORF also happen in the poly-purine sequences, which may induce small increase of ribosomal collision. This is a somewhat surprising finding given that the conventional model of translation elongation does not involve the regulation by eIF4A, yet the authors used simple and elegant bioinformatic analyses to show an intriguing new role of a key translation initiation factor. I like the novelty and implication of this study, however many analyses need more details or control.

Specific concerns:

1. Fig. 1B , 1C and 1E. Their data showed that the enrichment of RPF mainly happen in the first 75 codon of ORFs, however the reason seems unclear. Could this because the sequence bias, i.e., if the 5' end of ORF have more enriched with poly-purine motifs? They should look at such bias, especially in the context of codon usage (i.e., if the AG rich codon were used more in the first 75 codons).

2. Fig. 2A and 2B, because these two plots use different scale of y-axis, it is actually confusing at the first glance, because we expect to see more difference in RUG rather than RDG. They should somehow emphasize that the disome density is much smaller for RDG, and thus the seemingly larger difference in RDG (Fig 2A) is less relevant. They may want to combine these data into a single plot (with 4 curves). Also in this fig2B, there is a peculiar periodic distribution of disome in RUGs after the RocA03 treatment. The author should give an explanation (or speculation) on this.

3. In their experiments with translation reporters (Fig. 5), they only used the IRES-dependent translation, which account for translation of a small number of mRNAs. However the majority of RDG and RUG probably do not contain IRESs, and thus the author may also want to test their model using the translation reporter genes with cap-dependent initiation.

4. The translation reporters used in Fig. 5 contain many G rich sequences that may form the G-quadruplex structure. Have the author test the structures of this region and see if there is G-quadruplex that might affect translation elongation?

5. Fig. 6, it is nice that they used the data in cancer cells to examine the functional relevance of such regulation in cancer cells. Since many cancer cells also have proteomic data, I am wondering if they could look into the proteomic data and see the translation efficiency of the genes affected by RocA at the elongation or initiation step.

Responses to the Reviewers' Comments

We would like to thank the reviewers for their time and efforts in reviewing our manuscript. We are glad to see that all the three reviewers showed great interests in our study and provided highly positive feedbacks. The reviewers have made very thoughtful suggestions and comments, which have greatly helped us further improve the manuscript. We now have added a series of analyses and more discussions to address the questions. Please refer to the following point-by-point responses for the details.

Reviewer #1 (Remarks to the Author):

Rocaglates have attracted great interest due to their potential for tumor treatment and anti-COVID19 effects. Re-analyzing published data from Iwasaki and Ingolia's group with the novel disome profiling data of their own, Li et al. elegantly showed that rocaglates may inhibit the translation elongation. The mechanistic detail should be a splendid addition as a novel mode of translation repression by the leading compounds. This reviewer recommended addressing the following issues before the publication.

Major point:

1. This reviewer is deeply concerned about the authors' interpretation of ribosome footprints in 5' UTR. In the earlier paper (Iwasaki et al. Nature 2016, Wolfe et al. Nature 2014), those reads were attributed to "80S" ribosomes on upstream ORF (uORF), but not scanning 40S ribosomes, since 40S footprints were hard to recover without crosslinking as shown in TCP-Seq (Archer et al. Nature 2016 and other many manuscripts). Since authors referred to the footprints in 5' UTR as 40S footprints, the authors should rewrite the manuscript all through the manuscript.

Response: We are sorry for not making this clear and for causing this misunderstanding. We did not interpret the RPF reads in 5' UTR as 40S footprints. We totally agree with the reviewer and the previous papers that the ribosome footprints are from 80S ribosomes (Iwasaki et al., 2016; Wolfe et al., 2014). As reported by Iwasaki et al., clamping of eIF4A by RocA on the polypurine sequences in 5' UTRs results in blockage of 43S scanning, which has been shown to enhance uORF translation, thereby generating 80S footprints enriched in 5' UTR. We have revised the text at multiple places of the manuscript to clarify the potential misunderstanding (pages 4, 5, 20). Nevertheless, our study is focused on the shifted ribosome footprint distributions in CDS upon RocA treatments.

2. Related to the point above, the enhanced uORF translation by rocaglates has been suggested as an additional parameter to promote translational repression at the initiation step, but not the primary reason (Iwasaki et al. Nature 2016). Clamping of eIF4A to polypurine motif (this effect per se could not be monitored by ribosome profiling) should be what happens first (then this may lead to uORF translation but not always). Thus, 80S reads on 5' UTR should be a direct indicator of rocaglate sensitivity in translation initiation.

Response: We thank the reviewer for the notes. As discussed above, we totally agree with the reviewer on the view of the 80S footprints in 5' UTR as an indicator of translation initiation blockage induced by rocaglate. We have revised the text in the manuscript to make this clear (pages 4, 5, 20).

3. The nomenclature of "RPF upregulated" gene (and also "RPF downregulated" gene) should

be carefully used. Rocaglates (or other translation inhibitors too) definitely leads to global translation repression (even for the RUGs). Deep sequencing is always “relative” unless spike-in controls are added to library preparation. Thus, the correct wording here should be less sensitive/high sensitive or equivalent.

Response: We agree that the previous nomenclature of RUG and RDG could be misleading. We thank the reviewer for the suggestion. However, in our opinion, these two sets of genes were perturbed by RocA at different stages of translation, and therefore, their sensitivities to RocA per se may not be directly comparable. It may also be misleading to name them as less or high sensitive genes, as they could be equally sensitive in terms of the overall translation inhibition by RocA. We suggest to name these two sets of genes as elongation repressed genes (ERGs) and initiation repressed genes (IRGs).

4. Regarding the short 5' UTR genes, the authors classified them as TISU which does not require eIF4A and scanning. However, currently, this model is challenged by the report of Gu et al. Nat Commun 2021 that showed the backward (3'-to-5') scanning. The authors carefully characterized those mRNAs, scanning independency, and the interpretation of the data.

Response: We thank the reviewer for pointing this out. Indeed, the study by Gu, et al. has shown that eIF4A could promote bi-directional PIC scanning for start codon selection (Gu et al., 2021). This study has provided an operational mechanism for translation initiation of the mRNAs with ultra-short 5' UTR. However, it is also well demonstrated that the mRNAs with short 5' UTR tend to have very low translation efficiency unless they bear TISU (Elfakess et al., 2011). This earlier study demonstrated that such TISU-dependent translation fits in the cap-dependent slot-in model of translation initiation with minimal scanning, which is agreed by the paper of Gu et al. Nat Commun 2021. Therefore, based on these prior studies, although we cannot completely rule out the backward scanning, it is disputable whether such eIF4A-dependent bi-directional scanning is a dominant factor in determining the translations of the mRNAs with TISU elements. Nevertheless, our study is focused on the ribosome occupation patterns during translation elongation perturbed by RocA, whereas the uni- or bi-directional scanning affects translation initiation. We have reworded the text and added new discussion in the manuscript (page 21).

5. Figs 3 and 4 indicated the important parameters (charge, hydrophobicity, and polypurine motif) that are associated with ribosome stalling by rocaglates. The authors should clarify which factor is the primary one and which is additive to enhance the primary effect. Detailed data analysis and/or reporter assay may be helpful to dissect the role of these parameters.

Response: This is indeed an intriguing question. As shown by Fig. 3, in general, the ERGs and the IRGs exhibit differential codon usages, charges, and hydrophobicity, suggesting that compared to the IRGs, the ERGs are subjected to sub-optimal translation elongation. However, this does not explain why the ERGs and the IRGs respond to RocA differently. Therefore, in Fig. 4, we further showed that the ribosome stalling events upon RocA treatment take place mainly in the upstream of poly-purine sequences of the ERGs, an effect being lost under the context eIF4A mutation. This strongly indicates that the poly-purine sequences are the primary factors that drive the elongation inhibitory effect of RocA in a manner depending on eIF4A.

Here, to further confirm this conclusion, we analyzed the sequence features of the regions enriched by ribosome footprints (RocA vs DMSO ≥ 1.5 fold) within the first 50 codons of the ERGs upon 0.3 μ M RocA treatment. The positions with depleted ribosome footprints in the regions (RocA vs DMSO ≤ 0.67 fold) serve as references for comparison. No significant difference of the local cAI, tAI, charge, and hydrophobicity was observed between these two sets of regions in CDS of the ERGs (Fig. S11, reproduced as follows). Therefore, the features

of translation elongation, such as codon usage, charge, and hydrophobicity, do not seem to determine the inhibitory effects of RocA on translation elongation. They rather provide a suboptimal overall context for slow translation elongation, in which ribosome stalling due to eIF4A binding to the poly-purine motifs in response to RocA would be more likely to be observed. These results have been added in the revised manuscript (page 15)

Fig. S11. Sequence features of RocA-enriched or -depleted positions

6. Although the authors nicely showed that eIF4A clamping on CDS by rocaglates by re-analysis of iCLIP data, biochemical validation should be helpful. This could be conducted by toe printing assay of eIF4A with in vitro translation system or equivalent.

Response: We thank the reviewer for the suggestion. Indeed, MS2-GFP-RIP assays with the CDS regions of 3 ERGs confirmed the binding of eIF4A to these coding sequences (Fig. S6). Importantly, treatment of RocA at 0.3 μM resulted in further increase of the binding affinity (Fig. S6).

7. Given that polypurine in 5' UTR could evoke translation repression at the initiation step too, the authors should evaluate the efficacy of translation repression at the initiation step and elongation step and address which one is more dominant. This could be addressed by introducing synthetic polypurine motifs in 5' UTR or CDS in reporters, comparing the potency in repression by rocaglates.

Response: As suggested by the reviewer, we constructed luciferase reporter assays with synthetic polypurine motifs in 5' UTR or CDS. As shown in Figure 5B (reproduced as follows), RocA treatment induced similar efficacy of translation repression for the reporters with polypurine motifs found in CDS of 4 ERGs and for the reporters with polypurine motifs in 5' UTRs of 3 IRGs (Fig. 5B). However, although RocA induced similar extent of translation repression for these 7 reporters as examples, given the large numbers of ERGs and IRGs, we prefer not to make a generalized claim about the efficacy of RocA-induced translation

repression at the initiation or elongation stage.

Fig. 5B. Luciferase reporter assays with polypurine motifs in 5' UTR or CDS.

Minor point:

1. Although authors mentioned about the ribosome collision and the subsequent RSR/RQC pathway activation in the discussion section, they noted that this may not be led because of the high sensitivity of related components in translation. This was a contradiction.

Response: We are sorry for not making this clear. We showed that the key components in the RQC pathway that are responsible for ribosome splitting were largely inhibited by RocA (Fig. S13). Due to such defection of the RQC pathway, the stalled and collided ribosomes would not be efficiently resolved. As discussed in the last two paragraphs of the Discussion section (page 21-22), according to the recent studies (Vind et al., 2020; Wu et al., 2020), these un-resolved ribosomes may cause the RSR and could be a platform to recruit ZAK α and activate the MAPK signal pathway. We have revised the text of the manuscript to better convey this proposed machinery (page 21-22).

2. Related to the point above, in addition to ribosome stalling, rocaglates should suppress ribosome load by translation initiation inhibition. This should reduce the probability of ribosome collision (which is naturally taken by eIF2 α phosphorylation by RSR and eIF4E2-mediated repression in RQC). Authors should provide experimental evidence that rocaglate leads to RSR/RQC. Otherwise, authors should avoid the discussion regarding RSR/RQC.

Response: We thank the reviewer for the suggestion. Indeed, when the stress is not too severe, ribosome collision and stalling in the CDS regions could be resolved by eIF2 α phosphorylation by RSR (Wu et al., 2020) or via the GIGYF2-4EHP (eIF4E2) axis (Hickey et al., 2020; Sinha et al., 2020). However, it has been reported that RocA would not induce the phosphorylation of eIF2 α (Iwasaki et al., 2016), and GIGYF2 appears to be repressed by RocA (Fig. S13A). Considering that the key components of RQC were inhibited by RocA, we suspected that the RQC would not work under RocA treatment, leading to unresolved ribosome collisions in CDSs of ERGs. In addition, the downstream RSR/RQC pathways involve complicated processes and feedbacks, which are not solely dependent on the ribosome stalling in response to RocA. Simply testing some of the RSR/RQC genes is not sufficient to resolve the questions here, which are beyond the scope of the current study. Therefore, we do agree with the reviewer that more experimental investigations are needed to fully elucidate the responses and functions of the RSR/RQC pathways. To make the study more concise, we took the reviewer's suggestion and removed the small subsection in the results that discusses about the RQC genes. A brief discussion is retained in the Discussion section, in order to raise this intriguing question for potential future research (page 21-22).

3. For Fig. 4E, blue line (for Aglaiaicized-DMSO) was hard to see. Please consider better coloring of the graph.

Response: We thank the reviewer for pointing this out. The two lines for Aglaiaicized-DMSO and Naïve DMSO largely overlap. To make the lines clearer, we changed the colors and set them partially transparent (Fig. 4E in the revised manuscript).

4. Please double-check the following figure citation in the manuscript.

Fig. 5E in line 20 at p12 should be Fig. 4E;

Fig. 5A in line 4 at p15 should be Fig. 5C;

Fig. 5B in line5 at p15 should be Fig. 5D

Response: We are sorry for these errors, and we thank the reviewer for pointing them out. All the citations have been double-checked and corrected.

5. For Fig. 5, please highlight which IRES was used for the experiments and the mechanism of the IRES in the main text.

Response: We used the HCV-like IRES in the present study, which was obtained from previous studies using the similar reporter assays (Iwasaki et al., 2016). We have included this information and the rationale of using IRES for the reporter assays in the main text (page 16). The HCV-like IRES, which could initiate translation by binding to the solvent side of 40S ribosomes, was used for translation initiation of the firefly and Renilla luciferase reporters. Such IRES-dependent and cap-independent translation initiation is independent of translation initiation factor eIF4A1 (Kwan and Thompson, 2018) and shown to be unaffected by RocA (Iwasaki et al., 2016).

6. Regarding Fig. 4C (and S6), authors should consider showing the same plots for CDS for RDGs and 5' UTR for RUGs, for control.

Response: We thank the reviewer for the suggestion. We have added the ribosome densities around the polypurine motifs in both the CDS of RDGs (IRGs) and 5' UTR of RUGs (ERGs) as references (Fig. 4C, shown below). There is no significant accumulation of ribosome footprints around the polypurine motifs in the 5' UTR of ERGs upon RocA treatments, possibly because of the short 5' UTRs. In fact, there are very few poly-purine motifs found in the ERG 5' UTRs for the analysis above.

By contrast, RocA induced significant ribosome stalling in the upstream of the polypurine motifs of the CDS of IRGs (Fig. 4C), but to much weaker extents compared to the ribosome stalling in the 5' UTR of IRGs. This is not surprising. Although it is well acknowledged that RocA clamps eIF4A onto poly-purine motifs in the 5' UTR (Iwasaki et al., 2016), this machinery would certainly not completely shut down translation initiation of all the mRNA transcripts of IRGs in every single cell. Thus, the ribosomes that successfully passed the start codons could be also subjected to blockage due to clamping of eIF4A on the CDS regions, like what we have observations for the ERGs.

Fig. 4C. Enrichment ratio of ribosome footprints around poly-purine motifs based on comparisons between RocAs and DMSO.

7. Regarding Fig. 4E, the authors should consider drawing the similar meta-analysis around as shown in Fig. 4C and 4D.

Response: We thank the reviewer for the suggestion. Indeed, the events of ribosome stalling induced by RocA in the upstream region of the poly-purine motifs in the 5' UTR of IRGs and CDSs of ERGs were almost completely lost under the context of eIF4A mutation (Fig. 4F, shown below). This analysis nicely supports our proposed machinery of RocA inducing ribosome stalling during translation elongation in an eIF4A-dependent manner.

Fig. 4F. Enrichment ratio of ribosome density around poly-purine motifs based on comparisons between RocAs and DMSO in HEK293 cells with EIF4A1 double mutations.

8. To confirm that RUGs are translationally repressed even though their footprint change looks like translation upregulation, western blotting of the endogenous targets of rocaglates would be appreciated.

Response: We thank the reviewer for the suggestion. However, given the long half-life of proteins (median half-life about 7 hours), translation inhibition by RocA for such a short period (30 minutes) would not result in reduction of protein abundance that is detectable by regular

mass-spectrometry or western blotting. On the other hand, prolonged treatment of RocA would result in more secondary effects, including transcriptional repression of genes (Santagata et al., 2013) in addition to the dual-modal repression of translation, which would make it difficult, if not impossible, to evaluate the changes of protein abundance due to repression of translation. Therefore, we prefer not to use western blotting assays for assessment of such acute translation inhibition here.

9. This reviewer recommends the authors examine the positional effect of the polypurine motif in CDS. More pronounced results may be obtained in motif enrichment (Fig. 4A-B) and footprint accumulation around polypurine motif (Fig. C-D) when focusing on CDS near from start, compared to the remaining parts.

Response: We thank the reviewer for the suggestion. We have added new analyses to address this question (Fig. S8, shown below). First, we compared the relative enrichment of the polypurine sequences (≥ 4 mer) in the first 75 codons and the remaining CDS sequences (Fig. S8A, B). The results showed that the polypurine sequences are slightly more enriched in the regions after the first 75 codons (Fig. S8A). The AG enriched codons do not show noticeable bias though (Fig. S8B). However, despite the fact that polypurine sequences are less enriched in the first 75 codons, the ribosomes are still more likely to be stalled, in response to RocA, at the first 75 codons compared to the remaining CDS regions (Fig. S8C-E). This is not surprising. As we are proposing in this study, clamping of eIF4A onto the poly-purine sequences in the 5' CDS regions induces ribosome stalling during early elongation, which should reduce the number of the ribosomes moving further down to the 3' CDS regions. In other words, repression of translation elongation naturally reduces the ribosome footprints in the down-stream CDS regions. By contrast, ribosome accumulation could still take place in the upstream regions of the RocA-induced eIF4A binding events. As a result, the metagene analysis of ribosome profiling, which measures the collective and averaged behavior of the transcripts pooled together, would show significant ribosome accumulation towards the start of CDS. The results and discussion above have been added into the revised manuscript (Fig. S8, page 12-13).

Fig. S8. Position effects of poly-purine motifs in the CDS of ERGs.

Reviewer #2 (Remarks to the Author):

Li, Fang, and Yu et al identify a new mode of translational inhibition by Rocaglamide A (RocA) in this work. Prior work found that RocA inhibits translation by binding to eIF4A and clamping it onto polypurine motifs in 5' UTRs. This work identifies a second mode of translational inhibition by RocA, which the authors propose is mediated by eIF4A-dependent interactions with polypurine motifs in early coding sequences. This finding is interesting and in general the experiments and analyses are performed well and this work should be published. Comments below seek to improve the work, with the comment about the mechanism the most important to address in my opinion.

Major comments:

If RocA is generally inducing a block in elongation by eIF4A clamping to polypurine motifs it is confusing why the accumulation of ribosomes is towards the start of the CDS rather than throughout the entire CDS. It would seem an elongating ribosome could be stalled by a clamped eIF4A anywhere in the CDS. An alternative explanation might be that stalls in scanning by eIF4A clamping promote initiation at upstream out of frame start codons, leading to accumulation of ribosomes early in the CDS until they encounter an out of frame stop codon. Prior work showed RocA causes changes to uORF initiation. Can the authors explore this possibility by analyzing out of frame start codons or the frame of ribosome protected footprints in RUGs or otherwise? Why else might ribosomes selectively accumulate near the start of the CDS? Is there a 5' bias in the distribution of polypurine motifs within CDSs? What would happen if the polypurine reporters were constructed with the polypurine in the middle or end of the CDS? Fully understanding this may be beyond the scope of this work and that's ok, but additional exploration seems warranted to better define the mechanism as distinct from an initiation defect.

Response: We thank the reviewer for the insightful comments. As stated below, we have added more analyses and discussion to address these intriguing questions.

1. Response to the comment about out-of-frame RPFs: As suggested by the reviewer, we analyzed the ribosome footprints from the 3 different open reading frames (frames 0, 1, 2) of RUGs and RDGs (renamed as ERGs and IRGs, respectively). Interestingly, as shown in the following figure, 3 μM of RocA, but not the two lower doses of 0.3 and 0.03 μM , induced increase of the RPFs in frame 1 for both the ERGs and IRGs. Therefore, as the reviewer has suspected, it is indeed possible that upon RocA treatments, severe ribosome stalling leads to upstream out-of-frame translation, whereas the relatively weaker stalling of translation does not lead to such a strong frameshifting. However, this analysis is still not conclusive enough, given the technical variations within the biological replicates of the control samples. Therefore, more extensive studies will be needed to precisely profile the out-of-frame translation as a potential secondary effect of RocA treatments.

2. Response to the position of RocA-induced ribosome stalling: We agree that according to the proposed mechanism, theoretically, RocA could induce and strengthen the binding of eIF4A to the polypurine motifs throughout mRNA transcripts. Furthermore, as discussed in the response to Reviewer 1's comment, the polypurine sequences are slightly more enriched in the regions after the first 75 codons (Fig. S8A, shown below). However, despite the fact that polypurine sequences are slightly less enriched in the first 75 codons, the ribosomes are still more likely to be stalled, in response to RocA, at the first 75 codons compared to the remaining CDS regions (Fig. S8C-E). This is not surprising. As we are proposing in this study, clamping of eIF4A onto the poly-purine sequences in the 5' CDS regions induces ribosome stalling during early elongation, which should reduce the number of the ribosomes moving further down to the 3' CDS regions. In other words, repression of translation elongation naturally reduces the ribosome footprints in the down-stream CDS regions. By contrast, ribosome accumulation could still take place in the upstream regions of the RocA-induced eIF4A binding events. As a result, the metagenome analysis of ribosome profiling, which measures the collective and averaged behavior of the transcripts pooled together, would show significant ribosome accumulation towards the start of CDS. The results and discussion above have been added into the revised manuscript (Fig. S8, page 12-13).

Fig. S8. Position effects of poly-purine motifs in the CDS of ERGs.

Indeed, in a similar context of translation elongation inhibition by heat shock treatment, we also see 5' bias of ribosome accumulation (Shalgi et al., 2013). In this case, ribosome accumulation and slowdown of translation elongation also alleviate further accumulation of ribosomes in the downstream CDS regions. Based on the discussions above, we suspect that the general 5' CDS bias of ribosome accumulation is unlikely to be due to potentially differential functions of the polypurine motifs at different CDS regions. Theoretically, as the reviewer has suggested, translation reporter assays with polypurine motifs at different positions would provide further insights into this question. However, such assays in our lab did not work, as changes of the

luciferase CDS or insertions of polypurine sequences into the luciferase CDS result in the protein products different than the wildtype luciferase. We have not been able to measure consistent and abundant luciferase activities from these largely mutated proteins.

Nevertheless, we agree with the reviewer that our results lead to some very intriguing questions, which are beyond the scope of this manuscript, but certainly are worth discussion and further investigation in the future with more extensive biochemical and molecular studies. We have modified the text and added further discussion about this point in the revised manuscript (page 12).

It would be helpful in the reporter experiments to also include canonical RocA 5' UTR clamping reporters. This would allow a side-by-side comparison of the magnitude of effect on 5' UTR clamping versus the new mode of RocA action identified in this work.

Response: We thank the reviewer for the suggestion. We have added canonical reporter assays with polypurine motifs in 5' UTR, which has been previously shown to be sensitive to RocA treatment (Iwasaki et al., 2016). The results are consistent to the published data. The side-by-side comparison shows that the magnitude of translation repression via the new mode of RocA by inhibiting translation elongation is similar to the translation repression due to RocA-induced 5' UTR clamping (Fig. 5B).

Fig. 5B. Luciferase reporter assays with polypurine motifs in 5' UTR or CDS.

The authors note changes in regulation to RQC-related genes, which is interesting. A component of the RQC involves mRNA degradation of target mRNAs. If the authors have RNA-seq data from these experiments, they could use a computational package such as INSPECT (PMID 25957348) to estimate changes in RNA stability. It would be interesting either way – perhaps RocA is inducing mRNA decay in RUGs, or perhaps the changes to RQC-related genes are blocking mRNA decay. If the authors do not have RNA-seq data, generating it is beyond the scope of this work in my opinion. If the authors wished, they could measure select transcript half-lives with EdU labeling or transcriptional shut-off, though this is also unnecessary.

Response: We thank the reviewer for the suggestion. Since we do not have the data of nascent RNA, we cannot easily infer the synthesis and decay of mRNA. However, as suggested by the reviewer, we calculated the ratio of pre-mature RNA to mature RNA (P/M) by applying INSPECT on the RNA-seq data. The results showed that the transcripts of ERGs and IRGs respond to RocA differently. In general, RocA treatments resulted in slightly upper shift of the P/M ratios of the IRG transcripts, indicating potential post-transcriptional decay of the mRNAs. The ERGs show a similar trend, albeit without statistical significance.

However, the results above are still preliminary. In the present study, we showed that the key components in the RQC pathway were largely inhibited by RocA (Fig. S13). This should result in defection of RQC, but it remains unclear whether such changes to RQC-related genes would induce mRNA decay as secondary effects of RocA-induced ribosome stalling. In addition, the downstream RSR/RQC pathways involve complicated processes and feedbacks, which are not solely dependent on the ribosome stalling in response to RocA. We do agree with the reviewer that more experimental investigations are needed to fully elucidate the responses and functions of the RQC pathways, which are worth further investigation in the future with more focused and systematic studies. To make the study more concise, we have followed the first reviewer's suggestion and removed the small subsection in the results that discusses about the RQC genes. A brief discussion is retained in the Discussion section, in order to raise this intriguing question for potential future research (page 21-22).

Minor comments:

It would increase the readability of the figures if the legends of RocA3, RocA03, and RocA003 were changed to 3 μM RocA, 0.3 μM RocA, and 0.03 μM RocA throughout the paper.

Response: We thank the reviewer for the suggestion. We have replaced the previous labels of RocA with actual concentrations in μM throughout the manuscript.

The reporter experimental format leaves open the possibility that the actual effect may be larger than is shown in Fig 5. Specifically, cells were transfected for 12 hours and RocA was only applied for 30 minutes. During this experiment, luciferase will accumulate prior to addition of RocA and translation will only change for 30 minutes. Is it possible to treat the cells for a longer duration with RocA? This may increase the relative impact of RocA on translation.

Response: We thank the reviewer for the suggestion. We have carefully tested longer treatment durations for RocA and other timepoints after transfection of the reporter plasmids. However, prolonged treatment of RocA would result in more secondary effects, including transcriptional repression of genes (Santagata et al., 2013) in addition to the dual-modal repression of translation, which would make it difficult, if not impossible, to evaluate the change of protein abundance simply due to repression of translation elongation. Therefore, after carefully comparing the time period of reporter plasmid transfection and RocA treatment, we picked the current settings for the reporter assays, which clearly illustrate the translation repression by RocA. However, we do agree with the reviewer that the actual impact of RocA on translation may be larger than it appeared in the assays.

Reviewer #3 (Remarks to the Author):

In the manuscript by Li et al, the authors reanalyzed a large number of previously published Ribo-profiling data, and found some new insight on the regulation of translation elongation.

Specifically, they found that treatment of Rocaglamide A (RocA), a natural product that selectively inhibit translation initiation by clamping eIF4A onto the poly-purine motifs of the 5'UTRs, can also increase the eIF4A occupancy on the coding region. They further found that such stall of eIF4A on ORF also happen in the poly-purine sequences, which may induce small increase of ribosomal collision. This is a somewhat surprising finding given that the conventional model of translation elongation does not involve the regulation by eIF4A, yet the authors used simple and elegant bioinformatic analyses to show an intriguing new role of a key translation initiation factor. I like the novelty and implication of this study, however many analyses need more details or control.

Specific concerns:

1. Fig. 1B , 1C and 1E. Their data showed that the enrichment of RPF mainly happen in the first 75 codon of ORFs, however the reason seems unclear. Could this because the sequence bias, i.e., if the 5' end of ORF have more enriched with poly-purine motifs? They should look at such bias, especially in the context of codon usage (i.e., if the AG rich codon were used more in the first 75 codons).

Response: We thank the reviewer for the suggestion. The other reviewers have raised the same question as well. As suggested, we compared the relative enrichment of the polypurine sequences (≥ 4 mer) in the first 75 codons and the remaining CDS sequences (Fig. S8). The results showed that the polypurine sequences are slightly more enriched in the regions after the first 75 codons (Fig. S8A). The AG enriched codons do not show noticeable bias though (Fig. S8B). However, despite the fact that polypurine sequences are less enriched in the first 75 codons, the ribosomes are still more likely to be stalled, in response to RocA, at the first 75 codons compared to the remaining CDS regions (Fig. S8C-E). This is not surprising. As we are proposing in this study, clamping of eIF4A onto the poly-purine sequences in the 5' CDS regions induces ribosome stalling during early elongation, which should reduce the number of the ribosomes moving further down to the 3' CDS regions. In other words, repression of translation elongation naturally reduces the ribosome footprints in the down-stream CDS regions. The results and discussion above have been added into the revised manuscript (Fig. S8, page 12-13).

Fig. S8. Position effects of poly-purine motifs in the CDS of ERGs.

2. Fig. 2A and 2B, because these two plots use different scale of y-axis, it is actually confusing at the first glance, because we expect to see more difference in RUG rather than RDG. They should somehow emphasize that the disome density is much smaller for RDG, and thus the seemingly larger difference in RDG (Fig 2A) is less relevant. They may want to combine these data into a single plot (with 4 curves). Also in this fig2B, there is a peculiar periodic distribution of disome in RUGs after the RocA03 treatment. The author should give an explanation (or speculation) on this.

Response: We thank the reviewer for the suggestion. We have merged these 4 curves into a

single plot. As the reviewer has anticipated, it is clearer that the disomes were increased in the CDSs of RUGs (renamed as ERGs), which supports our proposed effect of RocA in inducing ribosome stalling for ERGs. It is also noted that the disomes are decreased in the CDSs of RDGs (renamed as IRGs), which is also well expected due to the repressed translation initiation of IRGs in response to RocA.

Fig. 2A. Metagen plot showing the disome distribution of ERGs and IRGs.

The apparently peculiar periodic distribution of disome in ERGs is an artifact due to data fluctuation and low resolution of the X axis in the plot of Fig. 2. Autocorrelations of the disome densities in the CDSs of ERGs indicate that there is actually no periodicity.

3. In their experiments with translation reporters (Fig. 5), they only used the IRES-dependent translation, which account for translation of a small number of mRNAs. However the majority of RDG and RUG probably do not contain IRESs, and thus the author may also want to test their model using the translation reporter genes with cap-dependent initiation.

Response: We are sorry for not making this clear. IRES is known to initiate translation by binding to the solvent side of 40S ribosomes. Such IRES-dependent and cap-independent translation initiation does not require translation initiation factor eIF4A1 (Kwan and Thompson, 2018), and it is also shown to be unaffected by RocA (Iwasaki et al., 2016). In the present study, we obtained the HCV-like IRES reporter from previous studies using the similar assays (Iwasaki et al., 2016). Specifically, we simply used this reporter to preclude the potential effect of RocA on translation initiation. This allows us to verify the effect of RocA in inhibiting translation elongation, in addition to its canonical role in blocking the scanning-dependent translation initiation by promoting the binding of eIF4A1 to the polypurine motifs in 5' UTRs.

In the revised manuscript, we have added canonical cap-dependent translation initiation reporters, which bear polypurine motifs in the 5' UTR, as positive controls that are expected to be repressed by RocA via blockage of translation initiation (Iwasaki et al., 2016) (Fig. 5B).

4. The translation reporters used in Fig. 5 contain many G rich sequences that may form the G- quadruplex structure. Have the author test the structures of this region and see if there is G- quadruplex that might affect translation elongation?

Response: We have used the RNAfold software to search for G-quadruplexes in the poly-purine sequences in the reporter plasmids. The reporters do not have any G-quadruplex in the first 50-60 codons of the CDSs containing the poly-purine motifs. Indeed, G-quadruplexes share conserved sequence motifs, and they are not close to the G rich polypurine motifs in the reporters. In addition, the main conclusion of our study is that translation elongation can be repressed by RocA, an effect being mediated by RocA-induced eIF4A binding to the polypurine motifs. In other words, the selectivity of the RocA-induced translation elongation is attributed to the binding of eIF4A to its target polypurine motifs. G-quadruplex itself may inhibit translation elongation, but it is not related to eIF4A binding. We have added a brief note for the absence of G-quadruplexes in the reporters in the revised manuscript (page 16).

5. Fig. 6, it is nice that they used the data in cancer cells to examine the functional relevance of such regulation in cancer cells. Since many cancer cells also have proteomic data, I am wondering if they could look into the proteomic data and see the translation efficiency of the genes affected by RocA at the elongation or initiation step.

Response: We thank the reviewer for the insightful comment. However, we did not have the proteomic data in cancer cells treated by RocA. Besides, given the long half-life of proteins (median half-life about 7 hours), translation inhibition by RocA for such a short period (30 minutes) would not result in reduction of protein abundance that is detectable by regular mass-spectrometry or western blotting, especially for the proteins with high abundance such as ribosome proteins and mitochondrial proteins. On the other hand, prolonged treatment of RocA would result in more secondary effects, including transcriptional repression of genes (Santagata et al., 2013), which would make it difficult, if not impossible, to evaluate the change of protein abundance due to repression of translation. Therefore, we prefer not to use proteomic data for assessment of such acute translation inhibition here.

References

- Elfakess, R., Sinvani, H., Haimov, O., Svitkin, Y., Sonenberg, N., and Dikstein, R. (2011). Unique translation initiation of mRNAs-containing TISU element. *Nucleic Acids Res* 39, 7598-7609.
- Gu, Y., Mao, Y., Jia, L., Dong, L., and Qian, S.B. (2021). Bi-directional ribosome scanning controls the stringency of start codon selection. *Nat Commun* 12, 6604.
- Hickey, K.L., Dickson, K., Cogan, J.Z., Replogle, J.M., Schoof, M., D'Orazio, K.N., Sinha, N.K., Hussmann, J.A., Jost, M., Frost, A., et al. (2020). GIGYF2 and 4EHP Inhibit Translation Initiation of Defective Messenger RNAs to Assist Ribosome-Associated Quality Control. *Mol Cell* 79, 950-962 e956.
- Iwasaki, S., Floor, S.N., and Ingolia, N.T. (2016). Rocaglates convert DEAD-box protein eIF4A into a sequence-selective translational repressor. *Nature* 534, 558-561.
- Kwan, T., and Thompson, S.R. (2018). Noncanonical Translation Initiation in Eukaryotes. *Cold Spring Harb Perspect Biol*.
- Santagata, S., Mendillo, M.L., Tang, Y.C., Subramanian, A., Perley, C.C., Roche, S.P., Wong, B., Narayan, R., Kwon, H., Koeva, M., et al. (2013). Tight coordination of protein translation and HSF1 activation supports the anabolic malignant state. *Science* 341, 1238303.
- Shalgi, R., Hurt, J.A., Krykbaeva, I., Taipale, M., Lindquist, S., and Burge, C.B. (2013). Widespread regulation of translation by elongation pausing in heat shock. *Mol Cell* 49, 439-452.

Sinha, N.K., Ordureau, A., Best, K., Saba, J.A., Zinshteyn, B., Sundaramoorthy, E., Fulzele, A., Garshott, D.M., Denk, T., Thoms, M., *et al.* (2020). EDF1 coordinates cellular responses to ribosome collisions. *Elife* 9.

Vind, A.C., Snieckute, G., Blasius, M., Tiedje, C., Krogh, N., Bekker-Jensen, D.B., Andersen, K.L., Nordgaard, C., Tollenaere, M.A.X., Lund, A.H., *et al.* (2020). ZAKalpha Recognizes Stalled Ribosomes through Partially Redundant Sensor Domains. *Mol Cell* 78, 700-713 e707.

Wolfe, A.L., Singh, K., Zhong, Y., Drewe, P., Rajasekhar, V.K., Sanghvi, V.R., Mavrakis, K.J., Jiang, M., Roderick, J.E., Van der Meulen, J., *et al.* (2014). RNA G-quadruplexes cause eIF4A-dependent oncogene translation in cancer. *Nature* 513, 65-70.

Wu, C.C., Peterson, A., Zinshteyn, B., Regot, S., and Green, R. (2020). Ribosome Collisions Trigger General Stress Responses to Regulate Cell Fate. *Cell* 182, 404-416 e414.

REVIEWERS' COMMENTS

Reviewer #1 (Remarks to the Author):

This reviewer appreciates the authors' effort to improve the manuscript, addressing this reviewer's concerns. This reviewer recommended publishing this paper when the following minor points were clarified.

1. The experimental design of MS2-GFP-RIP assay (Fig. S6) (i.e., GFP-MS2-RNA mediated eIF4A pulldown) should be explained in the manuscript and or the schematic representation in the figure.
2. In the MS2-GFP-RIP assay (Fig. S6), the regions where eIF4A binds could not be specified in CDS, although the authors inserted CDS into the reporter mRNA. The corresponding sentences at lines 284 and 285 should be carefully revisited to avoid overstatement.

Reviewer #2 (Remarks to the Author):

The authors have responded to all reviewer comments adequately and the paper should be published. Thanks for your work.

Reviewer #3 (Remarks to the Author):

The author addressed most of my comments with some discussions and new analysis, most of which has also been addressed by revisions in the manuscript. However it seems that they did not make changes in the manuscript on some of the points, more specifically on the peculiar artifact related to the comment 2 and comment 5. I suggest them to directly discuss the potential artifact on Fig. 2b, and add some discussions/speculations related to the proteomic analysis in cancer cells.

For the comment 5, it is actually not too hard to conduct a MS analysis using cultured cancer cells treat with RocA, and they should be able to see some signals by comparing the proteins with short half-life. If they think it is too much to ask, they should at least give a detailed discussion.

Responses to the Reviewers' Comments

We would like to thank the reviewers for reviewing our revised manuscript again. We are glad that our responses to the comments have been well appreciated and that the reviewers support the publication of our manuscript. As suggested by the reviewers, we have added more descriptions of the experiments and discussions about our findings.

Reviewer #1 (Remarks to the Author):

This reviewer appreciates the authors' effort to improve the manuscript, addressing this reviewer's concerns. This reviewer recommended publishing this paper when the following minor points were clarified.

1. The experimental design of MS2-GFP-RIP assay (Fig. S6) (i.e., GFP-MS2-RNA mediated eIF4A pulldown) should be explained in the manuscript and or the schematic representation in the figure.

Response: We have added a schematic plot for the MS2-GFP-RIP assay in Fig. S6a. A brief description of this experiment has been provided in the main text (page 9).

2. In the MS2-GFP-RIP assay (Fig. S6), the regions where eIF4A binds could not be specified in CDS, although the authors inserted CDS into the reporter mRNA. The corresponding sentences at lines 284 and 285 should be carefully revisited to avoid overstatement.

Response: We thank the reviewer for this comment. We have reworded the sentences here to avoid overstatement.

Reviewer #2 (Remarks to the Author):

The authors have responded to all reviewer comments adequately and the paper should be published. Thanks for your work.

Reviewer #3 (Remarks to the Author):

The author addressed most of my comments with some discussions and new analysis, most of which has also been addressed by revisions in the manuscript. However it seems that they did not make changes in the manuscript on some of the points, more specifically on the peculiar artifact related to the comment 2 and comment 5. I suggest them to directly discuss the potential artifact on Fig. 2b, and add some discussions/speculations related to the proteomic analysis in cancer cells.

For the comment 5, it is actually not too hard to conduct a MS analysis using cultured cancer cells treat with RocA, and they should be able to see some signals by comparing the proteins with short half-life. If they think it is too much to ask, they should at least give a detailed discussion.

Response: We thank the reviewer for the suggestions. We have added a brief discussion about the apparent artifact in Fig. 2 (page 8). We have also provided discussions for not using the MS or WB assays to evaluate the inhibitory effects of RocA on translation in the results section (page 12).